# RUFY3 regulates endolysosomes perinuclear positioning, antigen presentation and migration in activated phagocytes

Rémy Char[1], Zhuangzhuang Liu[2], Cédric Jacqueline [3], Marion Davieau[3], Maria-Graciela Delgado[4], Clara Soufflet[1], Mathieu Fallet[1], Lionel Chasson[1], Raphael Chapuy[1], Voahirana Camosseto[1], Eva Strock[1], Rejane Rua[1], Catarina R. Almeida [5], Bing Su [6], Ana-Maria Lennon-Duménil[4], Beatrice Nal [1], Antoine Roquilly[3], Yinming Liang [2], Stéphane Méresse [1], Evelina Gatti[1,5] ✉ & Philippe Pierre [1,5,6] ✉

Endo-lysosomes transport along microtubules and clustering in the perinuclear area are two necessary steps for microbes to activate specialized phagocyte functions. We report that RUN and FYVE domain-containing protein 3 (RUFY3) exists as two alternative isoforms distinguishable by the presence of a C-terminal FYVE domain and by their affinity for phosphatidylinositol 3-phosphate on endosomal membranes. The FYVE domain-bearing isoform (iRUFY3) is preferentially expressed in primary immune cells and up-regulated upon activation by microbes and Interferons. iRUFY3 is necessary for ARL8b + /LAMP1+ endo-lysosomes positioning in the pericentriolar organelles cloud of LPS-activated macrophages. We show that iRUFY3 controls macrophages migration, MHC II presentation and responses to Interferon-γ, while being important for intracellular *Salmonella* replication. Specific inactivation of *rufy3* in phagocytes leads to aggravated pathologies in mouse upon LPS injection or bacterial pneumonia. This study highlights the role of iRUFY3 in controlling endo-lysosomal dynamics, which contributes to phagocyte activation and immune response regulation.

Endosomes and lysosomes are cell organelles that primarily mediate uptake, degradation and recycling of materials delivered by endocytosis or autophagy[1]. As late endosomes, lysosomes, and other specialized lysosomal-like compartments share many membrane proteins (such as LAMP1 or LAMP2), we will refer to these compartments as endo-lysosomes (ELs). ELs participate in many cellular processes, including the regulation of cell metabolism or migration[1]. ELs also have cell-specific functions, such as antigen processing and presentation in professional antigen presenting cells (APCs) like macrophages or dendritic cells (DCs)[2]. These functions are highly dependent on EL's ability to move throughout the cytoplasm and concentrate as a cloud of membrane organelles in the pericentriolar area of the cell[3,4]. In this cloud, proteolytic compartments have propensity to tether and fuse together to facilitate membrane dynamics and material exchanges with other organelles, like the endoplasmic reticulum or the mitochondria[5,6]. The formation of the perinuclear organelles cloud is

[1]Aix Marseille Université, CNRS, INSERM, CIML, 13288 Marseille, cedex 9, France. [2]School of Laboratory Medicine, Xinxiang Medical University, Xinxiang, PR China. [3]Nantes Université, CHU Nantes, INSERM, Center for Research in Transplantation and Translational Immunology, UMR1064, F-44000 Nantes, France. [4]INSERM U932, Institut Curie, ANR-10-IDEX-0001-02 PSL* and ANR-11-LABX-0043 Paris, France. [5]Institute of Biomedicine (iBiMED), Department of Medical Sciences, University of Aveiro, 3810–193 Aveiro, Portugal. [6]Shanghai Institute of Immunology, Department of Microbiology and Immunology, Shanghai Jiao Tong University School of Medicine, Shanghai 200025, PR China. ✉e-mail: gatti@ciml.univ-mrs.fr; pierre@ciml.univ-mrs.fr

therefore key to maintain cell homeostasis and fulfill specialized functions, like activation by microbe-associated molecular patterns (MAMPs) and efficient antigen presentation or cytokine secretion by APCs[7,8]. Other external cues such as nutrient abundance and/or growth factors also induce cellular responses by altering EL spatial and temporal distribution[9]. Small GTPases like ARL8 or RAB7 have been shown to coordinate bidirectional (anterograde and retrograde) transport of EL along microtubules[10–12], through the regulated recruitment and/or activation of different effector proteins[13,14]. In addition to classical motor proteins, several effectors have been identified to regulate the localization and function of lysosomes via mammalian ARL8, including the homotypic fusion and protein sorting (HOPS)-tethering complex (HOPS)[12], Pleckstrin Homology and RUN Domain Containing M1 (PLEKHM1)[15] and PLEKHM2 (also known as SKIP)[11,16]. The interaction of ARL8-GTP with HOPS could promote late endosomes-lysosomes fusion by tethering lysosomes with Rab7/Rab2a+ late endosomes in cooperation with PLEKHM1 and PLEKHM2[15,17]. Perinuclear clustering of EL promote fusion with autophagosomes, which in turn leads to the lysosome-mediated mTORC1 signaling and the activation of autophagy[18]. Conversely, growth factors trigger anterograde lysosome movement toward the cell periphery, activating the mTORC1 signaling required for protein synthesis and autophagy inhibition[4]. Nutrient-driven lysosome positioning is also a critical determinant for ER remodeling and focal adhesion disassembly at the cell membrane[19,20]. Hence, bidirectional lysosome movement and distribution are key aspects of cellular adaptation to environmental cues. However, the exact molecular mechanisms linking lysosomes to the anterograde or retrograde transport systems are yet to be fully elucidated.

Recently the RUN and FYVE domain-containing protein 3 (RUFY3), has also been shown to be an ARL8b effector[21,22]. The RUN and FYVE domain-containing protein (RUFY) family encompasses five conserved genes displaying relatively broad tissue expression. The different RUFY proteins have been described to regulate endosomal trafficking, autophagy and cell migration[23]. They share a common structural organization with an N-terminal RUN domain, several coiled-coil (CC) motifs and a phosphatidylinositol 3-phosphate (PtdIns(3)P)-interacting C-terminal FYVE domain. Distinct from other RUFY proteins, RUFY3 was originally described to lack part of the C-terminal CC2 domain and the entire FYVE domain[24]. The interaction of RUFY3 with the filamentous actin network through a complex formed together with Rap2 and Fascin 1 (FSCN1) is critical for axonogenesis and growth cone development[25–27]. Expression of *rufy3* transcriptional spliced variants can however be detected in other tissues and cell types, with one uncharacterized mRNA variant coding for a predicted C-terminal region extended by 200 amino acids and containing a putative FYVE domain with potential affinity for PtdIns(3)P[23]. This variant of RUFY3 (RUFY3XL)[23] was recently shown to promote coupling of ELs to dynein-dynactin and regulate their trafficking along microtubules[21,22]. We report here that this larger FYVE domain-bearing isoform of RUFY3 is principally expressed in immune cells, including DCs and macrophages, and was termed "iRUFY3" in order to distinguish it from its previously characterized shorter and neuron-specific isoform (nRUFY3). We show that iRUFY3 expression and function are regulated by MAMPs, type-I or type II Interferons (IFN), as well as upon nutrient starvation. We confirm that in activated macrophages, like in HeLa cells[21,22], iRUFY3 drives the pericentriolar clustering of ARL8b/LAMP1+ ELs. In line with its association to ARL8b, targeted deletion of *rufy3* inhibits the formation of *Salmonella enterica*-containing vacuoles and replication in macrophages. Intriguingly, in *rufy3*-/- phagocytes, the lack of EL perinuclear clustering potentiates Interferon-γ-dependent antigen processing and MHC II-restricted presentation in vitro, while also affecting cell migration. In vivo, this translates into a pro-inflammatory state leading to an aggravated inflammatory syndrome after LPS injection or bacterial pneumonia. iRUFY3 appears therefore as a novel ARL8b/PtdIns(3)P effector required to regulate spatio-temporal EL positioning in the pericentriolar organelle cloud to coordinate antigen presentation and inflammatory function of MAMPs activated phagocytes[2].

## Results

### The FYVE domain bearing RUFY3 isoform is preferentially expressed in activated immune cells

We identified RUFY3 and RUFY4, as two proteins highly expressed in bone marrow-derived dendritic cells (bmDCs), likely involved in the regulation of endosome and autophagosome dynamics in response to Interleukin-4 (IL-4) exposure or MAMPs detection[28,29]. Genomic databases interrogation indicates that the *rufy3* mRNA is submitted to tissue dependent alternative splicing with 7 different transcripts detected in mouse. Along the previously characterized *rufy3* mRNA coding for a protein of 53 kDa (469 aa, nRUFY3, NP_001276705.1, transcript 4), we identified a larger isoform coding for a protein of 74.8 kDa (iRUFY3/RUFY3XL, 669 aa, NP_001276703.1, transcript 1) displaying a C-terminal extension of 200 amino acids containing a putative FYVE domain[23] (Fig. 1a). According to the ImmGen expression atlas[30] and qPCR monitoring, mouse *irufy3* (transcript 1) and *nrufy3* (transcript 4) mRNAs expressions are mutually exclusive with *nrufy3* isoform restricted to brain and *irufy3* mostly detected in immune cells (Fig. 1b and Supplementary Fig. 1a-c). Tissue analysis by immunoblot confirmed this restricted expression pattern, with spleen, lymph nodes, lung and thymus expressing solely iRUFY3 (Fig. 1c and Supplementary Fig. 1c) and meningeal samples displaying the two isoforms, being enriched both in neuronal cells and macrophages[31]. Only traces of RUFY3 isoforms were detected in other organs like liver or kidney (Supplementary Fig. 1d). Selective isoform expression was confirmed by immunoblot (Fig. 1d) and qPCR in Raw267.4 macrophages (RAW), MuTu DC[32], GM-CSF- or FLT3L-differentiated bmDCs (GM-CSF-DC or Flt3L-DC) (Supplementary Fig. 1a-c). The *rufy3* gene was inactivated using CRISPR/Cas9 technology in RAW macrophages (*rufy3*-/-, Supplementary Fig. 1e), prior reconstitution with a myc-tagged version of each isoform (RAW-iRUFY3 and RAW-nRUFY3) and comparative analysis of their respective expression (Fig. 1d). As inferred by the prediction of several coil-coiled domains in its secondary structure (Fig. 1a) and irrespective of the LPS activation state, iRUFY3 was found to form a dimer of around 150 kDa after PAGE in non-denaturing conditions (Fig.1e). Upon Flt3L-DC, MuTu DC, bone-derived or alveolar macrophages activation with different MAMPs, *rufy3* was the only *rufy* family member to be up-regulated transcriptionally after 8 h of stimulation (Fig. 1f and Supplementary Fig. 1b and 1f-g). A similar observation was made upon type-I IFN stimulation, with a near to 6-fold increase in *rufy3* transcription over 6 h (Fig. 1g). Expression of the iRUFY3 protein followed the same trend in MAMPs-activated Flt3-DC (Fig. 1h). Differently from primary cells, LPS activated-RAW did not up-regulate *rufy3* mRNA and exhibited a limited loss of protein expression (Fig. 1i), suggesting that transformation might affect some aspect of *rufy3* transcriptional regulation and homeostasis.

### iRUFY3 is recruited to perinuclear LAMP1 + ELs upon LPS activation or nutrient starvation

iRUFY3 is equipped with a putative FYVE domain likely to interact with phosphatidylinositol-phosphate enriched membrane domains. Although predicted to form two consensual "zing finger" motifs, this domain however lacks the tandem histidine residue cluster that defines affinity for PtdIns(3)P in other identified FYVE motifs (R + **HH**C + xCG)[33,34] (Fig. 1a). With regard to their affinity for PtdIns(3)P, FYVE domain-containing proteins generally regulate endosomal membrane traffic and are mostly found associated to endo-lysosomes, phagosomes and forming autophagosomes[34]. We examined by confocal microscopy the localization of RUFY3 in *rufy3*-/- RAW macrophages stably reconstituted with individual myc-tagged isoforms of the molecule focussing on the endocytic pathway at steady state, upon

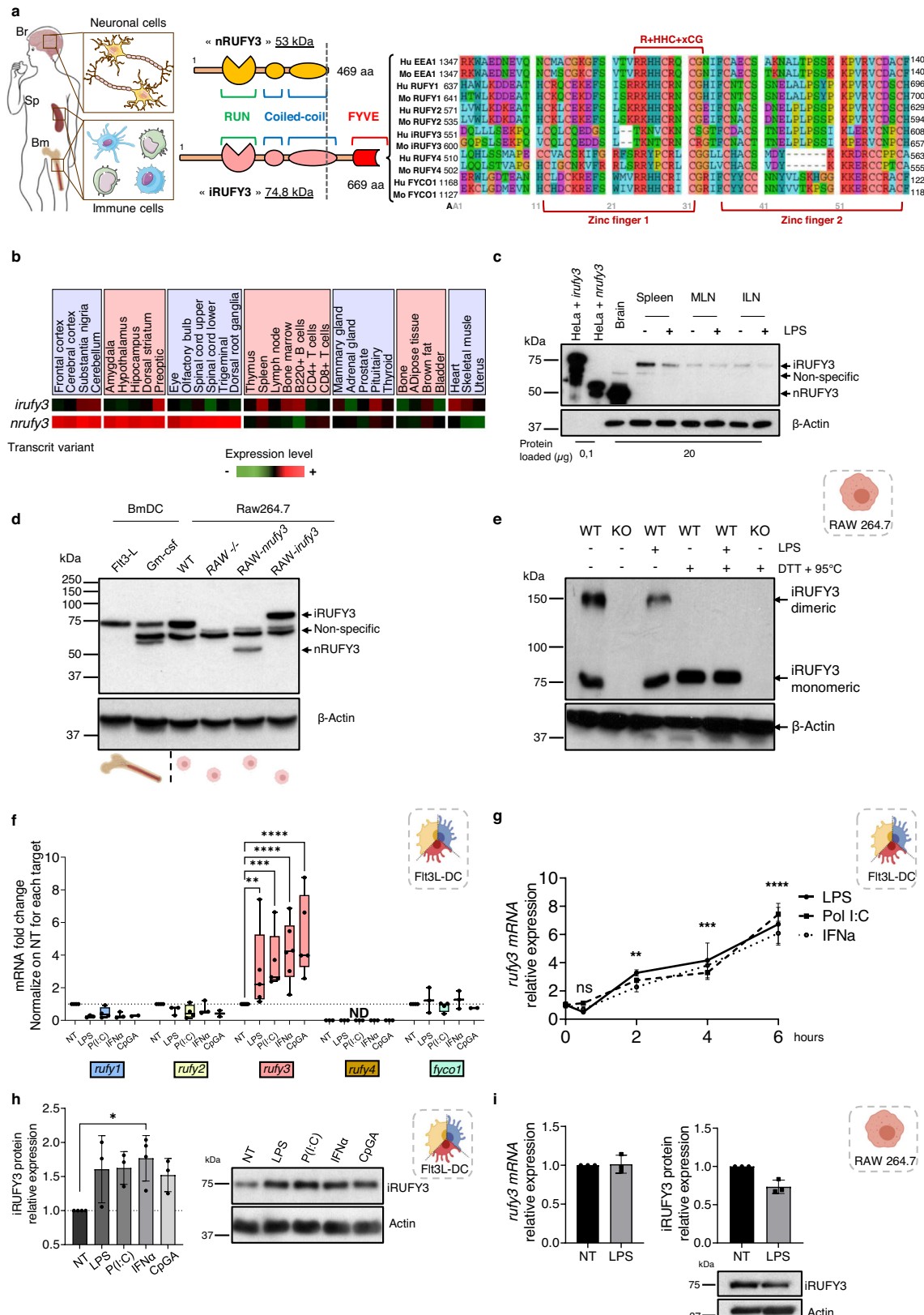

LPS activation or nutrient starvation (EBSS). iRUFY3 distributed mostly with concentrated perinuclear organelles, while nRUFY3 was clearly diffused and cytosolic (Figs. 2a and 3a). The sorting endosomes (SE) markers EEA1 and Syntaxin 6 did not show any significant distribution overlap with iRUFY3, conversely to the late endosomal marker LAMP1, that displayed extended co-localization with iRUFY3 (arrows, Fig. 2b–d). As expected, RAB11A+ recycling endosomes (RE) were also distributed in the perinuclear area, and therefore in close vicinity to RUFY3, however no co-localization between the two molecules could be observed after careful examination (arrows, Fig. 2b–d). Pearson's and Manders' Overlap Coefficient (MOC) quantification both indicated that iRUFY3 recruitment to LAMP1+ ELs was augmented in LPS-

**Fig. 1 | The FYVE domain bearing RUFY3 isoform is preferentially expressed in activated immune cells. a** Schematic representation of RUFY3 functional domains organization. RUN: RPIP8, UNC-14 and NESCA domain, CC1: coiled-coil 1 domain, CC2: coiled-coil 2 domain, FYVE: Fab1, YOTB, Vac1 and EEA1 domain. An amino acids alignment of the iRUFY3 FYVE domain with other known FYVE domains is displayed showing the absence of normally conserved Histidine tandem from the iRUFY3 and RUFY4 Zinc finger 1 domains. Color code: cyan for hydrophobic positions (A, V, I, L, M), turquoise for aromatic positions (F, Y, W, H), red for basic residues (K, R), purple for acidic residues (D, E), green for non-polar charged (N, Q, S, T), salmon for cysteine (C), orange for glycine (G) and yellow for proline (P). Br Brain, Sp Spleen and Bm Bone marrow. **b** Comparative expression of the different *rufy3* isoforms in mouse tissues (RNA seq Immgen database). The shorter *rufy3* transcript (*nrufy3*) is mostly expressed in brain, while the FYVE-bearing larger transcript (*irufy3*) is enriched in bone marrow, lymphoid organs and immune cells. **c, d** Immunoblot showing the expression of RUFY3 protein isoforms in brain, spleen, mesenteric (MLN) and iliac (ILN) lymph nodes with or without LPS stimulation in vivo (**c**) and (**d**) in bone marrow-derived DCs and Raw264.7 (RAW) macrophages in vitro. Actin is not detectable in HeLa i/nRUFY3 over-expressing control due to minimal sample

loading (100 ng). Blots are representative of two independent experiments. **e** Immunoblot after non-denaturing PAGE revealing iRUFY3 dimerization in RAW Wild Type (WT) and *rufy3* knock out (KO). Blot is representative of two independent experiments. **f, g** Quantification by RT-qPCR of the *rufy* genes family transcripts in Flt3L-bmDCs after 6 h exposure to TLR ligands and IFN-α (**f**) and of *rufy3* over 6 h (**g**). For **f**, the boxplot data represent medians, interquartile ranges and spikes to upper and lower adjacent values. Each dot represents one independent experiment. Statistical significance was established using two-way ANOVA with Dunnett's multiple comparisons test. For **g**, each dot represents one independent experiment ($n = 3$) and data are presented as mean values +/- SD. Statistical significance was established using one-way ANOVA with Tukey's multiple comparisons test. **h, i** Immunoblot detection of iRUFY3 levels in Flt3L-bmDCs (**h**), and (**i**) quantification by RT-qPCR of the *rufy3* mRNA and iRUFY3 protein by immunoblot in RAW stimulated or not with LPS. All data are presented as mean values +/− SD. $n = 3$ independent experiments except for IFNα condition where $n = 4$. Statistical significance was established using one-way ANOVA with Dunnett's multiple comparisons test. For all panels (*$p < 0,05$; **$p < 0,01$; ***$p < 0,001$; ****$p < 0.0001$).

activated and EBSS-starved cells, in which EL clustering was increased (Fig. 2c–d and Fig. 3a), as further evidenced after 3D rendering (Fig. 3b and 3c) and localization at the microtubule organization center (MTOC, Fig. 3d). Differently from EBSS starvation, LAMP1 overlap with RUFY3, which was increased in LPS-stimulated cells compared to non-treated conditions, mostly concerned pericentriolar ELs and not all LAMP1 + EL subsets (MOC, Fig. 2c), suggesting some level of selectivity for RUFY3 recruitment upon LPS-stimulation.

Importantly, reconstitution iRUFY3 levels in *rufy3*[-/-] cells did not impact EL organization compared to WT RAW, while FYVE-less nRUFY3 was never associated with LAMP1+ ELs in all the experimental conditions tested (Fig. 3a). iRUFY3 association to organelles was confirmed using subcellular fractionation, with a portion of iRUFY3 pelleting with membrane-enriched fractions (invariant chain/CD74 positive), including ELs (Fig. 3e). As anticipated from confocal imaging, the proportion of iRUFY3 associated with membranes was strongly increased after LPS treatment.

## iRUFY3 promotes LAMP1 + ELs clustering and associates with ARL8b

Previous observations involving RUFY3 in EL positioning[21,22], and RUFY3 recruitment to LAMP1+ organelles in activated phagocytes, led us to investigate whether iRUFY3 participates to EL clustering in conditions known to trigger microtubules (MT)−dependent retrograde transport like LPS exposure or nutrient starvation. This was supported by the redistribution of RUFY3 + /LAMP1+ organelles to the RAW cells periphery upon depolymerization of MT with nocodazole (Noc) (Supplementary Fig. 2a), improving the imaging resolution of distinct RUFY3 + /LAMP1+ ELs and demonstrating the importance of the MT network to support perinuclear clustering. Two methods for image analysis-based quantitation of LAMP1 + /LAMP2+ ELs clustering were developed using organelle to centroid distance calculation and monitoring of pixel distribution by 360° Angular Scanning monitoring Organelle Distribution (360-ASOD, see methods and Supplementary Fig. 2b). Both methods confirmed that RUFY3 deletion interferes strongly with MT-based LAMP + EL perinuclear positioning, with most organelles remaining in the cell periphery at steady state and upon starvation in RAW *rufy3*[-/-], as judged from the calculated LAMP1 clustering index as well as 360-ASOD polarity and variance distribution (Fig. 4a). Conversely, iRUFY3 ectopic expression in HeLa cells induced EL perinuclear clustering (Supplementary Fig. 2c), as previously observed for RUFY4[21,28].

As anticipated from the published ARL8 interactome analysis[21,22], iRUFY3 was found to co-colocalize with ARL8b at steady state and upon RAW cell activation by LPS or nutrient starvation (Fig. 4b). ARL8b distribution was strongly affected by RUFY3 deletion, remaining

associated with LAMP1+ ELs in the periphery and failing to reach the MTOC area, as indicated by a strong reduction of ARL8b clustering index in non-treated and EBSS starved RAW *rufy3* -/-. Colocalization and a possible physical interaction (distance <40 nm) between ARL8b and iRUFY3 was also observed by confocal microscopy in a proximity ligation assay (PLA) at steady state, the intensity of which was reinforced upon starvation and LPS stimulation (Fig. 4c). A reduction in the average distance among PLA spots also confirmed that in addition of being increased, ARL8b and RUFY3 site of interactions were also clustered upon LPS and EBSS treatments, as expected for LAMP1+ ELs (Fig. 4c) and suggesting that RUFY3 and ARL8b could be co-recruited to allow EL retrograde transport on microtubules[21,22]. These results are in line with the recent observations that ARL8b and RUFY3 physically interact upon ectopic overexpression in HeLa cells[21,22]. The same experiments performed in RAW-nRUFY3 did not indicate any interaction of nRUFY3 with ARL8b (Fig. 4c) confirming that the neuronal isoform does not localize to ELs (Supplementary Fig. 2a), and confirming that the domain of interaction of iRUFY3 with ELs is located within the last 200 aa residues of the protein and not within its RUN domain[21,22].

## Silencing of ARL8b alters iRUFY3 expression and endosomal localization

The role of ARL8b in recruiting iRUFY3 on LAMP1 + EL was examined after silencing Arl8b expression by RNAi. A bulk 40% decrease in ARL8b levels was reached after 24 h of silencing in RAW-iRUFY3 (Fig. 5a). Surprisingly, bulk levels of RUFY3 were also decreased by 20% upon ARL8b silencing, demonstrating a strong biochemical link between the two molecules, although a reciprocal effect on ARL8b expression upon loss of RUFY3 was not observed in *rufy3*[-/-] RAW (Fig. 5b). Imaging and cytofluorographic analysis confirmed an average loss of 50% in RUFY3 expression (MFI) in silenced cells. The magnitude of this loss at the single cell level was strongly correlated to the intensity of ARL8b down-modulation (Supplementary Fig. 3a). However, although dimmed by silencing, remaining ARL8b and RUFY3 expression levels were sufficient to perform quantitative imaging analysis (Fig. 5c and Supplementary Fig.3a and 3b). The down-modulation of ARL8b notably impacted the dynamic of ELs, resulting in reduced LAMP1 staining intensity during EBSS treatment. iRUFY3 recruitment to LAMP1+ organelles was also reduced upon ARL8b silencing, as indicated by MOC calculation (Fig. 5c). This was confirmed by the analysis of voxel gated co-localization channels (Supplementary Fig. 3b), where a decreased number of LAMP1+ compartments remained RUFY3 positive in the same conditions (Fig. 5c). Interestingly, although the knock-down of ARL8b was not fully efficient in most cells, the distribution of RUFY3 remained pericentriolar, yet

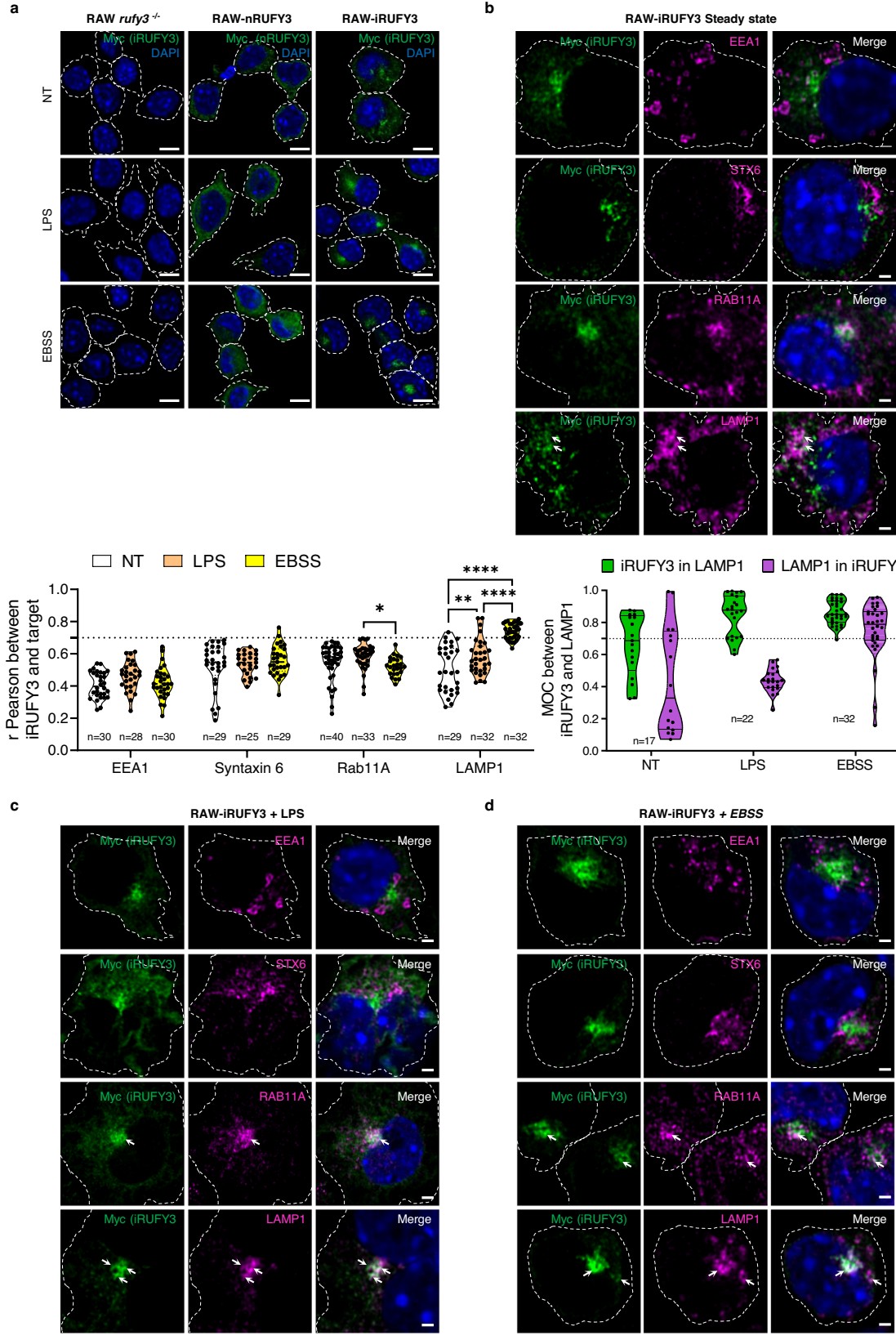

distinct from ARL8b+ structures (MOC Fig. 5c, and Supplementary Fig. 3a-b). In silenced cells, RUFY3 was predominantly cytosolic, but interestingly also associated with some ARL8b-negative endo-lysosomes. Thus, although its recruitment to ELs appears to be mostly linked to that of ARL8b, RUFY3 may have the capacity to interact with other membrane-associated molecules if ARL8b levels are decreased.

Given the presence of a FYVE domain in C-terminus of iRUFY3, we evaluated the importance of PtdIns(3)P in its recruitment to ELs. iRUFY3 distribution was investigated after treatment with the FYVE finger containing phosphoinositide kinase (PIKfyve) inhibitor, YM201636. PIKfyve is a phosphatidylinositol lipid kinase, which binds and transforms PtdIns(3)P in PtdIns(3,5)P, and plays a critical role in

**Fig. 2 | iRUFY3 co-localizes with perinuclear LAMP1+ endosomes upon LPS activation or nutrients starvation. a** Airyscan Immunofluorescence confocal microscopy (AICM) panels showing iRUFY3 intracellular distribution in RAW *rufy3*[-/-] cells stably expressing a control empty vector (left, RAW *rufy3*[-/-]), *nRUFY3* (middle, RAW-nRUFY3) and *iRUFY3* (right, RAW-iRUFY3) at steady state (top), after 16 h of LPS stimulation (middle) and 6 h of nutrient starvation (EBSS, bottom), dotted lines indicate cell boundaries. Scale bar is 5 μm. These results are representatives of *n* = 3 independent experiments with >100 cells observed by experiments. **b**–**d** AICM images of RAW-iRUFY3 showing RUFY3 intracellular (myc) distribution compared to endocytic markers EEA1, Rab11A, Syntaxin-6 (STX6) and LAMP1 at steady state (**b**), after 16 h LPS stimulation (100 ng/mL) (**c**) or 6 h starvation (EBSS) (**d**). White arrows indicate co-localization (LAMP1) or absence of co-localization (Rab11a) with RUFY3. Pearson co-localization and Mander's overlap coefficients (MOC) were calculated using Image J for b-d panels. A highly significant co-localization score is considered above 0.7 for Pearson's. Statistical significance was established using two-way ANOVA with Tukey's multiple comparisons test (*$p < 0,05$; **$p < 0.01$; ****$p < 0,0001$). Each dot represents the mean off all Z-stack from one region of interest. Scale bar 1 μm.

endosomal membrane trafficking[35]. PIKfyve inhibition with YM201636 causes abnormal accumulation of PtdIns(3)P and results in the appearance of enlarged vacuoles in mammalian cells due to alterations in membrane fission and formation of tubules[36,37]. YM201636 treatment clearly altered RUFY3 distribution in control cells, exacerbating the loss of RUFY3 and ARL8b co-localization, as well as their association with LAMP1+ organelles in ARL8b-silenced RAWs-iRUFY3 (MOC, Fig. 5c). The fact that RUFY3 remained associated to membranes upon YM201636 treatment, suggests that an excess of PtdIns(3)P may synergize with ARL8b silencing to decrease RUFY3 recruitment to LAMP1+ ELs, and potentiate its binding to other organelles or membrane structures (Fig. 5c). An hypothesis strongly supported by the association of iRUFY3 with YM201636-induced LAMP1-negative tubules emanating from ELs in steady state conditions and more prominently in starved cells, thus competing with RUFY3 binding to LAMP1+ ELs (Supplementary Fig. 4a-c). YM201636 treatment failed to alter nRUFY3 cytosolic distribution, both at steady-state and in starved cells (Supplementary Fig. 4c), further pointing at the critical role of iRUFY3 C-terminal domain in EL targeting.

## PtdIns(3)P contributes to iRUFY3 endosomal localization

iRUFY3 distribution was next investigated upon nutrient starvation (EBSS) and pharmacological inhibition of the Class III PtdIns(3)P-kinase VPS34[38] (Fig. 6). VPS34 inhibitor (VPS34i) interferes with the main pathway of PtdIns(3)P synthesis and affects the recruitment to endosomes of several endogenous PtdIns(3)P-binding proteins, including EEA1[39,40]. VPS34i treatment reduced the clustering of LAMP1 + EL in EBSS starved cells (Fig. 6a, b). The pericentriolar positioning of RUFY3 and the decoration of some tubular organelles was however not affected by VPS34i, although most of its co-localization with LAMP1 was lost, confirming that PtdIns(3)P generation next to ARL8b recruitment contributes to its interaction with ELs. iRUFY3 direct binding to PtdIns(3)P enriched domains was however not supported by co-localization experiments with GFP-2xFYVE (Fig. 6c). GFP-2xFYVE is a protein probe that specifically associates with PtdIns(3)P-enriched domains upon ectopic expression[41] and is sensitive to VPS34i (Fig. 6c). In the different conditions tested, iRUFY3 was never found co-localized with this probe (Fig. 6c), suggesting that although PtdIns(3)P contributes to iRUFY3 recruitment on EL membranes, it might be through an indirect process involving ARL8b or via an interaction with other types of PtdIns lipid enriched domains.

## RUFY3 is important for intracellular *Salmonella* replication in macrophages

To test functionally the impact of RUFY3 on EL function, we performed *Salmonella* infection, since ARL8b/PLEKHM2/HOPS- and Rab7/PLEKHM1-dependent EL mobilizations are particularly important for the maturation of *Salmonella*-containing vacuoles (SCV) and bacterial replication in macrophages[42–44]. Colony forming unit assay (CFU) monitoring indicated that *Salmonella* replication was reduced by 40% in infected *rufy3*[-/-] cells compared to WT RAW cells (Fig. 7a). Complementation of *rufy3*[-/-] cells with myc-iRUFY3 restored *Salmonella* replication to WT levels, while myc-nRUFY3 expression had little effect on rescuing CFU titers (Fig. 7a), in line with its lack of interaction with ARL8b and ELs. Equivalent uptake of WT and replication-incompetent

(Δ*sifA*) bacteria was observed after 2 h of infection in *rufy3*[-/-] and WT RAW cells, showing that iRUFY3 is not involved with bacterial phagocytosis (Fig. 7b), although It is found in the vicinity of some SCVs, together with LAMP1 and the bacterial effector PipB2[45] at 16 h post-infection (Fig. 7c and 7d). Experiments performed in EBSS conditions to enhance ELs perinuclear concentration, revealed that *Salmonella* infection fully prevents LAMP1 clustering presumably by recruiting RUFY3 to the SCV. Thus, iRUFY3 is therefore important to allow intracellular *Salmonella* replication in macrophages.

## iRUFY3, but not nRUFY3, is required for macrophages and DC migration

Controls of axonal growth depends on nRUFY3 interaction with actin filaments[27]. nRUFY3 is therefore key for nervous system development, remodeling and function, explaining the embryonic lethality displayed upon full *rufy3* genetic inactivation in mouse[46]. We thus generated a novel transgenic mouse with floxed alleles at the borders of the exon 3 of the *rufy3* gene (Supplementary Fig. 5a). Upon Cre recombinase expression, the deletion of exon 3 prevents expression of functional RUFY3. *Rufy3*[lox/lox] C57/BL6 mouse were crossed with a Itgax-cre deleter strain to specifically inactivate *Rufy3* in CD11c-expressing cells, including most DC subsets and alveolar macrophages[47]. Loss of *rufy3* was confirmed at the mRNA and protein levels by RT-PCR and immunoblot in CD11c+ splenocytes, Alveolar macrophages and GM-CSF-DCs (Supplementary Fig. 5b-e). We used microfabricated channels, that mimic the confined geometry of the interstitial space in tissues[48] to find that *rufy3*-deficient GM-CSF-DCs were unable to increase their migration speed in response to LPS (Fig. 8a). We confirmed this observation in *rufy3*[-/-] RAW cells, which display a strong reduction in their migratory properties in a scratch/wound healing assay in vitro (Fig. 8b). Complementation with iRUFY3 fully restored the migration capacity of resting and LPS-activated *rufy3*[-/-] cells, while nRUFY3 did not rescue this deficit (Fig. 8b). ELs positioning and dynamics is therefore critical for cell migration, by potentially acting on the recycling of integrins or the remodeling of focal adhesion dynamics, as shown for ARL8b-dependent anterograde transport[20], RAB7b and for PIKfyve activity[49]. Importantly, nRUFY3 was not able to rescue normal migration of RAW *rufy3*[-/-], although this FYVE-less protein is critical for neurons migration and axonogenesis[25,26]. This lack of redundancy confirms that the two RUFY3 isoforms operate in different molecular environments to perform distinct functions. The described interaction nRUFY3 with Rap2 and Fascin (FSCN1) in neurons[27], could be inexistant for iRUFY3 in macrophages due to the lack of *fscn1* expression in RAW cells (Fig. 8c) and other macrophage subsets[50]. iRUFY3-dependent ELs positioning seems therefore necessary for the migration of circulating and tissue resident MAMPs activated-phagocytes independently of its interaction with FSCN1.

## *Rufy3* deletion alters antigen processing and presentation in macrophages

We next examined the consequences of IFN-γ exposure on RAW macrophages to establish whether MHC II dynamics could be altered upon iRUFY3 inactivation in macrophages. Interestingly although iRUFY3 levels were weakly augmented by IFN-γ treatment (Fig. 9a), RAW *rufy3*[-/-] showed an exacerbated response to this cytokine with

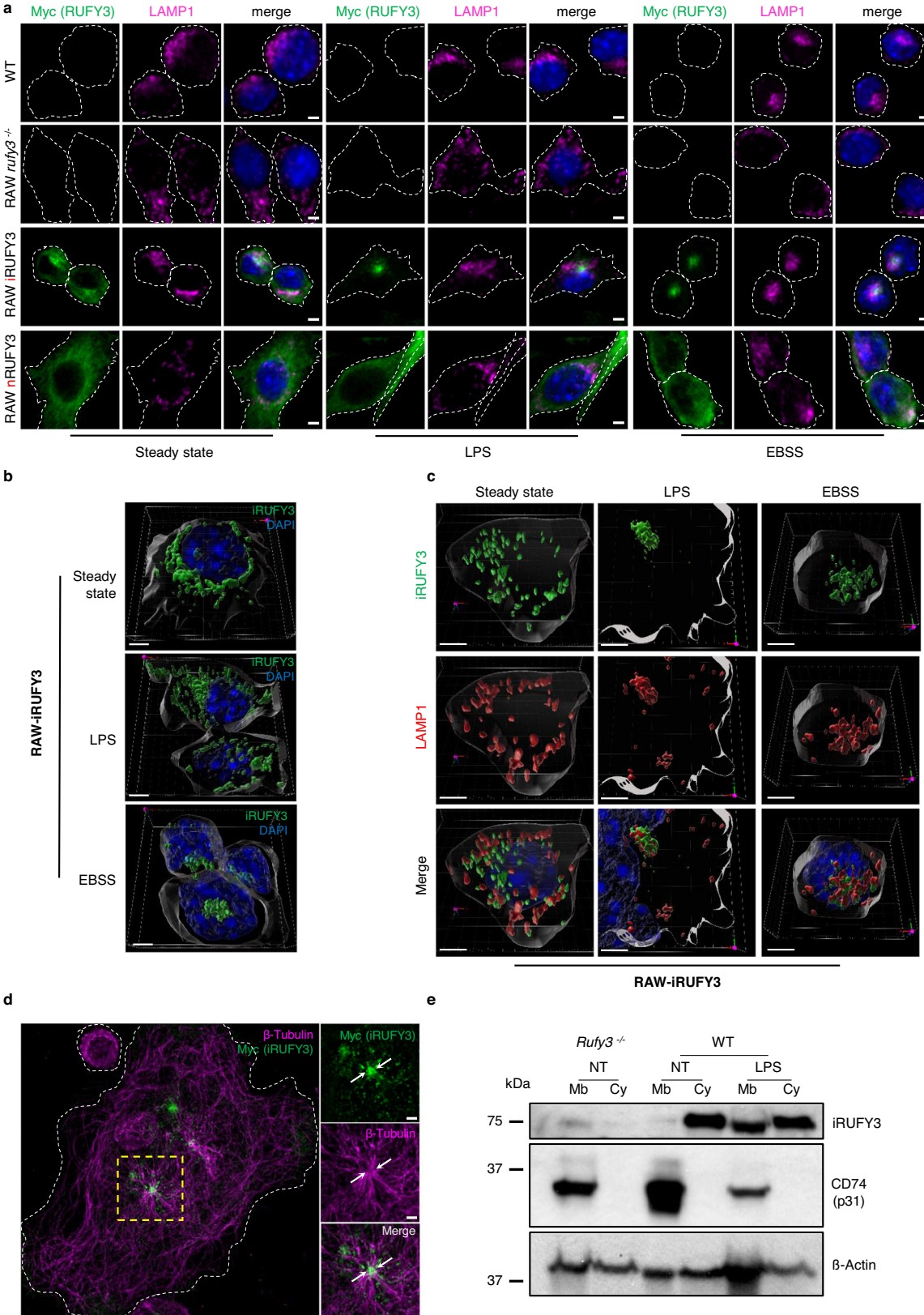

elevated levels of surface MHC II and numbers of cells expressing it (Fig. 9b and 9c). This elevation was also observed at the transcriptional level since induction of major histocompatibility complex II transactivator (CIITA) and I-Ad-alpha and beta MHC II mRNAs were augmented in absence of iRUFY3 (Fig. 9d), indicating that *rufy3* deletion potentially augments CIITA co-transcriptional activity downstream of

IFN-γR1[51]. Accumulation of MHC II molecules at the cell surface strongly increased in *rufy3* [-/-] cells exposed to IFN-γ for 24 h (Fig. 9e), suggesting that RUFY3 could also regulate MHC II transport and exogenous antigen presentation.

Given the capacity of IFN-γ to potentiate endosomal proteases activity[52], we monitored by flow cytometry the degradative capacity of

**Fig. 3 | iRUFY3 associates with perinuclear LAMP1+ endosomes upon LPS activation or nutrient starvation. a** AICM showing LAMP1+ vesicles and RUFY3 distribution in Raw264.7 macrophages (WT), *rufy3* depleted (*rufy3* -/-) and stably complemented with nRUFY3 or iRUFY3. Dotted lines indicate cell boundaries. Scale bar 2 μm. These results are representatives of n = 3 independent experiments with >100 cells observed by experiments. **b** 3D reconstruction showing iRUFY3 at steady state (top), after 16 h LPS stimulation (middle) or 6 h nutrient starvation (bottom). Scale bar 5 μm. **c** 3D display showing iRUFY3 intracellular distribution compared to LAMP1 at steady state, after 16 h LPS stimulation and 6 h starvation (EBSS). Scale bar 2 μm. *X, Y* and *Z*-axis are defined at the bottom right. **d** AICM showing iRUFY3 distribution and the microtubule network (ß-tubulin) in LPS-activated RAW-iRUFY3 cells. Scale bar 2 μm. This result is representatives of n = 2 independent experiments with >100 cells observed by experiments. **e** Analysis of cytosolic (Cy) and membrane (Mb) fractions from post-nuclear supernatants (PNS) of RAW and RAW *rufy3*-/-. After LPS treatment, RUFY3 is enriched in Mb fractions that contain the type-II transmembrane Invariant chain p31/CD74 used as fractionation control marker. This blot is representative of n = 2 independent experiments.

RAW cells, using DQ-ovalbumin, which is a self-quenched conjugate of ovalbumin that exhibits bright green fluorescence upon proteolytic degradation. Although similar to control cells at steady state, processing of DQ-ovalbumin was exacerbated in *rufy3* -/- macrophages treated with IFN-γ (Fig. 9f). Expression and proteolytical processing of the MHC II-associated Invariant chain (Ii, CD74) was also examined in presence of the cysteine protease inhibitor N-morpholinurea-leucine-homophenylalanine-vinylsulfone-phenyl (LHVS)[53] (Fig. 9g). LHVS treatment results in the accumulation of discrete Ii intermediate fragments known as p22 (22 kDa) and p10 (10 kDa), that remain associated to MHC II αβ complexes and mediate their retention in ELs[52,54]. Differently from MHC II, intact Ii isoforms (Ii-p31 and Ii-p41) expression was detected in non-stimulated RAWs and increased by IFN-γ stimulation (Fig. 9g). *Rufy3* -/- RAWs again displayed high sensitivity to IFN-γ stimulation, with considerably increased levels of both Ii isoforms and accumulation of Ii-p10 over control cells. LHVS forced also a much greater accumulation (2-3 fold) of Ii-p10 and Ii-p22 in *rufy3* -/- cells treated with IFN-γ, indicating that both invariant chain synthesis and its endosomal processing are greatly accelerated in absence of RUFY3. This global enhancement in endosomal proteolysis and Ii-MHC II complexes maturation was translated into a nearly 2-folds enhancement of ovalbumin antigen presentation to DO.11 T cells by IFN-γ-treated *rufy3* -/- macrophages compared to control (Fig. 9h). Thus, although EL dynamic is globally slowed-down by RUFY3 inactivation, some key functions like IFN-γ signaling, as well as, MHC II-restricted antigen processing and presentation are enhanced in activated *rufy3* -/- RAW cells.

## *Rufy3* deletion in the CD11c+ cell compartment is pro-inflammatory

EL perinuclear positioning in response to MAMPs is a hallmark of DCs activation/maturation[8,55,56], which are characterized by higher surface levels of MHC II and CD86, like IFN-γ-treated macrophages. Global immunophenotyping of control and *rufy3lox/lox*-Itgax-cre mice, indicated the presence of proportionally higher numbers of activated (mature) cDC1 and cDC2 in the spleens of knock-out animals (Fig. 10a), compatible with a reduced egress of activated cells from the spleen in absence of functional iRUFY3. Upon intraperitoneal injection of LPS, a light splenomegaly (Fig. 10a and 10b) with abnormal accumulation of macrophages, pDC, mature cDC2, as well as NK cells was observed in the *rufy3lox/lox*-Itgax-cre mice (Fig. 10c and Supplementary Fig. 6a). *Rufy3lox/lox*-Itgax-cre animals were particularly sensitive to low doses of LPS (1.5 ng/g) and displayed abnormal immune cell infiltrations in the lung, which were characterized as B and T cells accumulating in tertiary lymphoid structures using specific B220 and CD3 staining (Supplementary Fig. 6b and 6c).

This pro-inflammatory phenotype, prompted us into investigating further how *rufy3* deletion in CD11c+ alveolar macrophages, which monitor the luminal surface of the epithelium where air-borne bacteria grow[57], affects the response to lung infection. We infected intra-tracheally *rufy3lox/lox*-Itgax-cre mice and control litter mates with fluorescent YFP expressing *E. coli* to cause a primary pneumonia[58] and evaluate disease progression and the associated immune response at day 3 and 7 post-infection (Fig. 10d). Clinical signs of pneumonia, analysed by measuring overall survival, weight loss, tolerance to pain,

were found to be more severe in *rufy3lox/lox*-Itgax-cre deficient animals at day 3 post-infection (Fig. 10e). However, this increased severity did not prevent, nor delay, the overall recovery observed at day 7. RUFY3 deficiency in lung phagocytes impacts therefore only transiently the response to bacterial infection. This was in line with the equivalent levels of bacterial phagocytosis measured in the different lung macrophage subsets irrespective of their genetic background (Supplementary Fig. 7a and 7b). We next examined the numbers and phenotypes of macrophage subsets and lymphocytes present in the lungs at steady state and during infection (Fig. 10f and Supplementary Fig. 7a). Seven days post-infection, alveolar macrophages (AM) were 2-fold more numerous in *rufy3*-deficient animals than control, with a greater proportion of IFN-γ-producing cells. Higher levels of surface MHC II and of the IFN-γ inducible co-stimulatory receptor CD48[59] were also observed in rufy3 -/- AM (Fig. 10f), but not in other CD11c-negative MACs subsets (I-MAC) (Supplementary Fig. 7c). Importantly, preceding the phenotype observed with AM, the numbers and proportion of IFN-γ-producing NK cells were significantly augmented 3 days post-infection in *rufy3lox/lox*-Itgax-cre animals, while levels of CD3 + T cells were reduced, particularly in the effector/memory T cells compartment (TMEM) (Fig. 10f). The proportion and capacity of CD8 + T cells to produce IFN-γ was however enhanced upon deletion of *rufy3* in the CD11c+ cells. This increased IFN-γ production and activated phenotypes of *rufy3* -/- AMs, strongly echoed with the sensitivity of *rufy3* -/- RAW to IFN-γ exposure in vitro, and was probably the cause of exacerbated inflammation in the challenged *rufy3lox/lox*-Itgax-cre mice. Overactivation of the T cells upon enhanced stimulation and IFN-γ exposure in the lungs of CD11c+ *rufy3*-deficient mouse might lead to rapid exhaustion, and premature contraction of the effector/memory compartment, as previously observed during infection[60].

## Discussion

We have identified an immune-cell specific RUFY3 splicing variant that contains a fully functional FYVE domain (iRUFY3). This discovery has now corrected the anomaly of including the shorter FYVE-less neuronal form[23] in the RUFY family[24]. Our research has shown that while PtdIns(3)P levels affect iRUFY3 recruitment to EL and other organelles, its non-fully consensual sequence seems to prevent direct binding to PtdIns(3)P-enriched membrane domains. RUFY3 is necessary for the transport of ARL8b + /LAMP1 + EL from the cell periphery to the perinuclear organelles cloud upon MAMPs detection or nutrient starvation. Its interaction with ARL8b, as well as the JIP4-dynein-dynactin complex in transfected HeLa cells[21,22], and on pericentriolar ARL8b+ ELs in activated macrophages (this study), suggests that RUFY3 acts as an effector of ARL8b to control microtubule-dependent retrograde transport of ELs[15] (Supplementary Fig. 8). RUFY3 genetic inactivation produces similar effects to that of ARL8b, including inhibition of *Salmonella* replication[61,62] and loss of migration capacity[20]. Interestingly, these phenotypes were previously linked to the ARL8/PLEKHM2 complex activity that controls kinesins-dependent MT plus end–directed transport of ELs, rather than dynein-mediated retrograde transport. Although RUFY3 seems to solely regulate ARL8b-dependent retrograde clustering of ELs, as suggested by its interaction with the JIP4-dynein complex[22], it might be also necessary to maintain EL

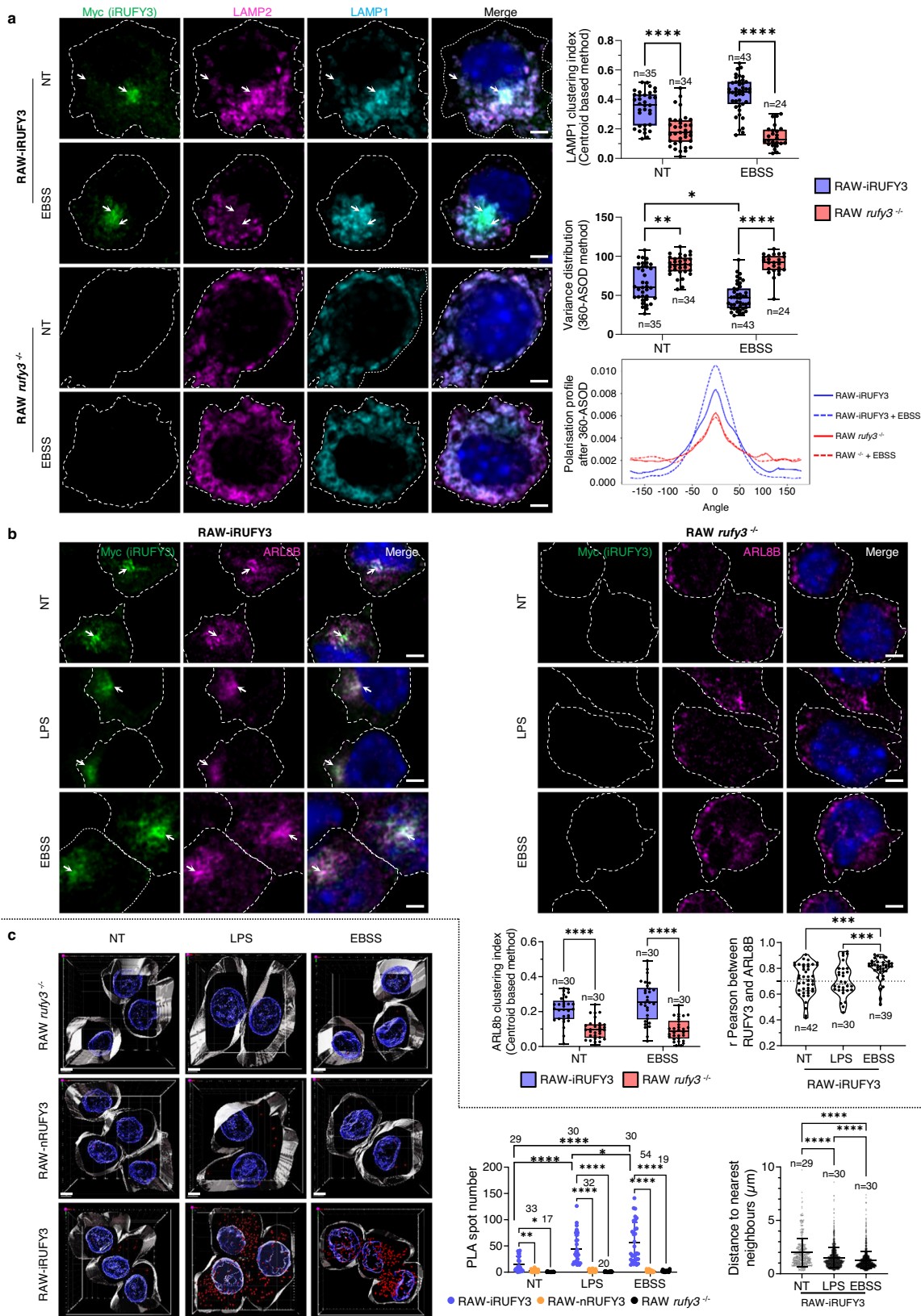

homeostasis and, indirectly regulate ARL8b-dependent antero-rograde transport.

The phenotypic differences reported in ARL8b-deficient cells compared to RUFY-deficient ones (this study), may be attributed to the restricted activity of RUFY3 on the clustering of a subset of ELs, conversely to the broader impact of ARL8b deficiency on both anterograde and retrograde traffic. It is worth noting that alternative pathways, such as the Rab7/RILP/dynein system, which also promotes the temporal and morphological compartmentalization of perinuclear ELs[3], may partially compensate for RUFY3 loss. However, the inter-pretation of these experiments may be complicated by the control exerted by ARL8b on RUFY3 expression. We have shown that silencing

**Fig. 4 | iRufy3 promotes LAMP1 + -lysosomes clustering and associates with ARL8b. a** Detection of myc-RUFY3 and LAMP1 + EL distribution by AICM in RAW-iRUFY3 quantified with centroid based method and 360-ASOD. Dotted lines indicate cell boundaries. Scale bar 2 μm. The boxplot data represent medians, inter-quartile ranges and spikes to upper and lower adjacent values. Statistical relevance was established using unpaired *t*-test (*$p < 0,05$; **$p < 0.01$; ***$p < 0.001$; ****$p < 0,0001$; ****$p < 0,0001$). **b** Distribution of iRUFY3 and ARL8b monitored by AICM in RAW-iRUFY3 and RAW *rufy3* [-/-] with centroid based method. Colocalization score and clustering index are shown. A clear co-localization score is considered above 0.7 for Pearson's. Scale bar 2 μm. The boxplot data represent medians, interquartile ranges and spikes to upper and lower adjacent values. Statistical relevance was calculated with one-way ANOVA with Tukey's multiple comparison test (****$p < 0,0001$; ****$p < 0,0001$). **c** Immunofluorescence Proximity Ligation Assay (PLA) was performed for iRUFY3 and ARL8b with 3D image reconstitution in RAW *rufy3* [-/-] (top), RAW-nRUFY3 (middle) and RAW-iRUFY3 (bottom) at steady state (left), 16 h LPS stimulation (center) or 6 h nutrient starvation (EBSS, right). Scale bar 5 μm. PLA spot quantification and distribution are shown. Clustering was quantified by nearest neighbor's distance calculation for each PLA spot. Data are presented as mean values +/- SD. For all panels numbers (*n*) of cells analyzed are indicated. Statistical relevance was calculated by Welch's *t*-test for clustering indexes, two-way ANOVA with Tukey's multiple comparisons test for co-localization and nearest neighbor distance (*$p < 0,05$; ***$p < 0.001$; ****$p < 0,0001$; ****$p < 0,0001$).

ARL8b in RAW-iRUFY3 causes a proportional loss of RUFY3 expression, which could be explained either by a potential instability of RUFY3 dimers when not efficiently recruited to ELs, and/or upon exclusion from a multimeric protein complex formed with ARL8b. Additionally, ARL8b silencing revealed interactions of iRUFY3 with other organelles, implying that it may have additional functions in membrane traffic than facilitating EL pericentriolar positioning. How bidirectional EL transport is regulated by ARL8b remains however unclear and it will be crucial to revisit this issue in the light of our findings on RUFY3 function. It is tempting to speculate that, similar to RAB7, ARL8b uses alternative recruitment of adapters like FYCO1 or RILP to determine the type of organelle and of molecular motors to partner with[3]. Interestingly, some of the adapter functions linking RUFY3 to ARL8b may be equivalent to that of FYCO1 for RAB7, as both molecules exhibit significant homology, and that FYCO1 can be classified as a RUFY family member (RUFY5)[23]. In this respect, the contribution of VPS34 or PIKfyve kinases and their responses to environmental cues are likely essential in fine-tuning RUFY3-dependent transport by defining the sites of PtdIns(3)P accumulation and tagging ELs for perinuclear transport and/or membrane exchanges with other compartments.

The exact nature of EL subsets requiring RUFY3 for their motility and the nature of the signal regulating these pathways will have to be further characterized. ELs are key organelles for nutrient sensing, processing and loading of antigens for presentation, as well as proteolytic activation or degradation of membrane receptors to induce or shut-down downstream signaling according to the circumstances[1,2]. Transport to the perinuclear organelles cloud is also necessary for antigens to reach MHC II molecules in the late endocytic compartments of D1 DCs, resulting in reduced antigen processing and presentation upon ARL8b silencing[7]. Interestingly, RUFY3's loss enhances antigen processing and presentation in IFN-γ–treated macrophages. This effect was confirmed in vivo by the pro-inflammatory phenotype of CD11c[cre]*rufy3* [fl/fl] mice displaying increased sensitivity to low dose of circulating LPS, as well as enhanced inflammation upon bacterial pneumonia. The stronger response to IFN-γ of RUFY3 [-/-] phagocytes could be a key determinant in promoting this situation, may be becoming dominant over moderate alterations of antigen processing and MHC II transport potentially caused directly by RUFY3 inactivation and EL mispositioning (Supplementary Fig. 8). Our efforts to investigate IFN-γR transport and signaling in *rufy3* [-/-] cells have been unsuccessful, due to the lack of appropriate probes[63]. However, a slowing-down of IFN-γR degradation and signaling activity upon reduction of EL distribution around the MTOC is a plausible hypothesis that will have to be evaluated.

Surprisingly, expression of the *rufy3* variants is mutually exclusive with the shorter form (nRUFY3) limited to neuronal tissues. To date, most of the data available on RUFY3, have been obtained in cells, in which solely nRUFY3 is expressed or induced, with most investigations focussed on neuronal polarity, cell migration or metastasis[27,46,64,65]. We have shown that solely iRUFY3 interacts with ARL8b and contributes to EL positioning upon cell activation. However, both isoforms contribute to cell migration, albeit in a non-redundant manner and probably amid

a cell-type specific biochemical context, like for instance FSCN1 expression. ARL8b and RUFY1 have also been shown to interact together on recycling endosomes and to regulate endosomes to TGN retrieval of CI-M6PR[66]. RUFY1 is required to control cell migration and invasion[23], suggesting that RUFY proteins might represent a family of adapter molecules specialized in specifying the direction of endosomes subsets transport, while controlling cell migration. It will be important to determine the contribution of the reduced migration observed in *rufy3* [-/-] phagocytes to the pro-inflammatory situation observed in CD11c[Cre]-*rufy3*[loxp/loxp] mice. Interestingly, it was recently shown that Rab7b regulates activated DC migration by linking lysosomes to the actomyosin cytoskeleton, which requires correct positioning to allow localized Ca$^{2+}$ release to activate myosin II and fast and persistent DC migration[48,67]. The preferential expression of iRUFY3 in immunocytes and its regulation by immune mediators, demonstrate, nevertheless, a key role for EL perinuclear positioning to modulate functions typically associated to APCs, including pathogen clearance, antigen presentation and inflammatory mediator production. These functions could be harnessed pharmacologically by interfering with the iRUFY3/ARL8b pathway.

## Methods

### Ethics statement
The research described in this manuscript complies with all relevant ethical regulations. Animal Studies were carried out in strict accordance with Guide for the Care and Use of Laboratory Animals of the European Union. All experiments were approved by the ethical committees PACA and PdL, under MESRI approval numbers APAFIS #18981-2019020710111763 and APAFIS #32506-2021072015469639. All efforts were made to minimize animal suffering.

### Mice
Wild-type (WT) female and male C57BL/6 mice were purchased from Janvier, France. *rufy3* [loxp/loxp] mice were developed at the Centre d'Immunophénomique (CIPHE, Marseille, France), for details see Supplementary Fig. 6a. Rufy3[loxp/loxp] were crossed with Itgax-Cre+ mice[47]. Mice were backcrossed to obtain stable homozygotic lines for the loxp sites expressing Cre. For all studies, age-matched WT and transgenic 6–12 weeks females were used. To compare with CD11c[Cre]-*rufy3* [loxp/loxp] mice, CD11c[Cre] mice littermates were used as control. All animals were maintained in the animal facility of CIML or CIPHE under specific pathogen–free conditions accredited by the French Ministry of Agriculture.

### Model of non-lethal acute bacterial pneumonia
Infection and analysis were performed as described[58]. Briefly, YFP-*E. coli* (strain DH5α with p-HG-1 plasmid) grown for 18 h in LB medium at 37 °C, were washed twice, diluted in sterile isotonic saline and calibrated by nephelometry. *E.coli* (75 μl, OD600 = 0.6−0.7) were injected intratracheally in anesthetized mice to induce a non-lethal acute pneumonia. Infected mice were monitored daily for weight loss and tolerance to pain. Immune cell purification from lungs at

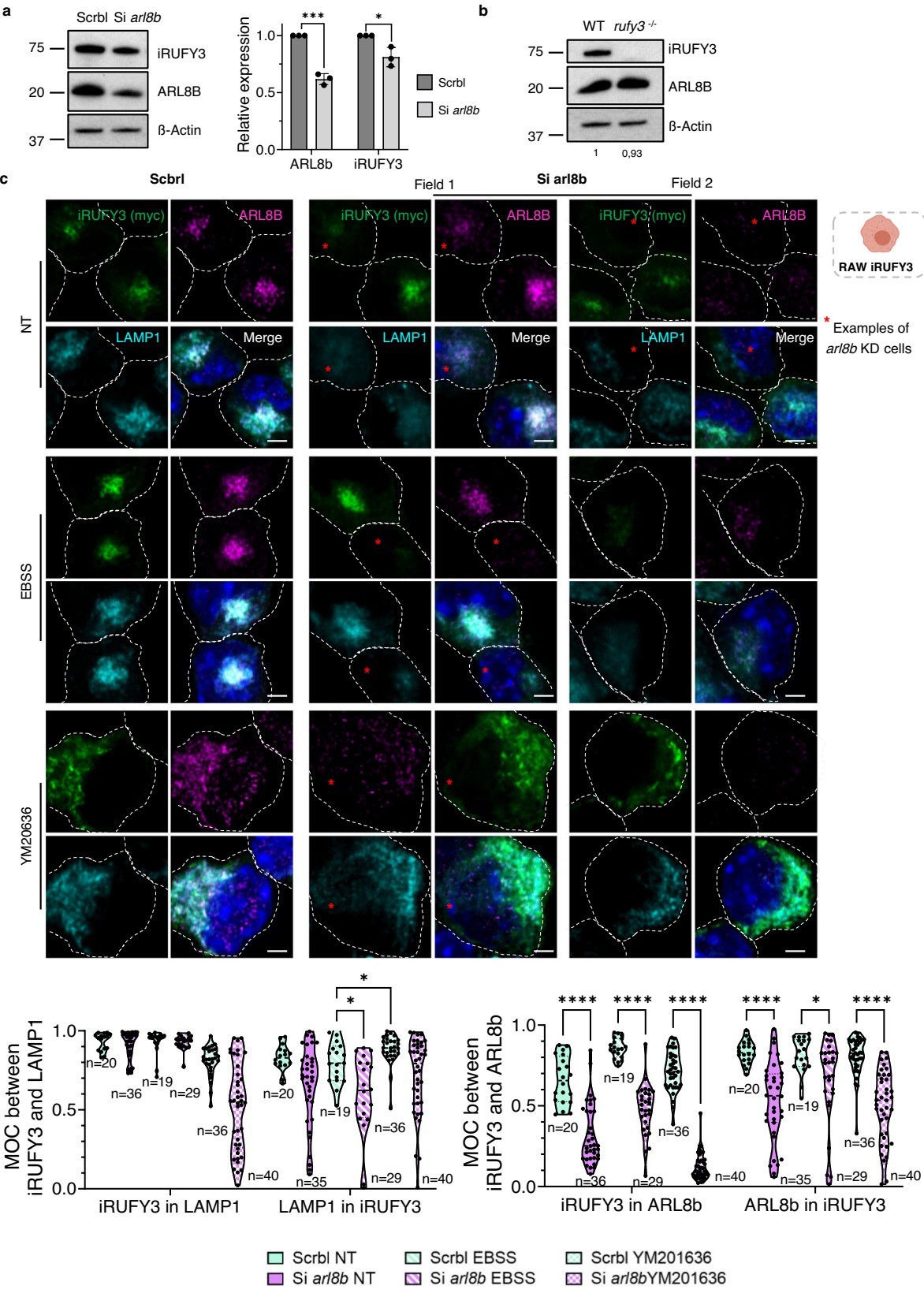

day 3 and 7 p.i., and analytical flow cytometry using conjugated monoclonal antibodies were performed as described[58]. After cell sorting from lung, over-activation with Phorbol Myristate Acetate (PMA, invivogen #tlr-pma) and Ionomycin (Invivogen, #inh-ion) solution was performed.

## Sequences alignment

*Rufy3* sequences from human and mouse were obtained from ImmGen and NCBI databases. Alignments were performed with Sea View analysis software V5.05[68]. For all alignments, amino acids are colored according to their biophysical properties. The accession

**Fig. 5 | Silencing of ARL8b alters iRUFY3 expression and endosomal localization. a** Immunoblot detection and quantification of iRUFY3 and ARL8b expression upon Arl8b RNAi silencing in RAW-iRUFY3. *N* = 3 independent experiments. Data are presented as mean values +/- SD. **b** Immunoblot detection of iRUFY3 and ARL8b expression in WT and *rufy3*⁻/⁻ RAW. The blot is representatives of two independent experiments. Number below actin represents ARL8B intensity normalized on WT. **c** Detection of iRUFY3, ARL8b and LAMP1 by AICM upon Arl8b silencing in RAW-iRUFY3 at steady state, upon starvation (EBSS, 6 h) or after treatment with PIKfyve

inhibitor (YM201636, 45 min, 5 μM). Dotted lines indicate cell boundaries and red stars indicate cells with particularly efficient ARL8b silencing. Scale bar is 1 μm. Mander's Overlay Coeficient (MOC) was calculated using Image J. One dot represents the mean of all z stacks from one cell. Statistical relevance was calculated by unpaired *t*-test (**a**) and two-way ANOVA with Tukey's multiple tests for co-localization (**c**). For (**c**) numbers (*n*) of cells analyzed are indicated. For all panels: *$p < 0.05$; **$p < 0.01$; ***$p < 0.001$; ****$p < 0.0001$.

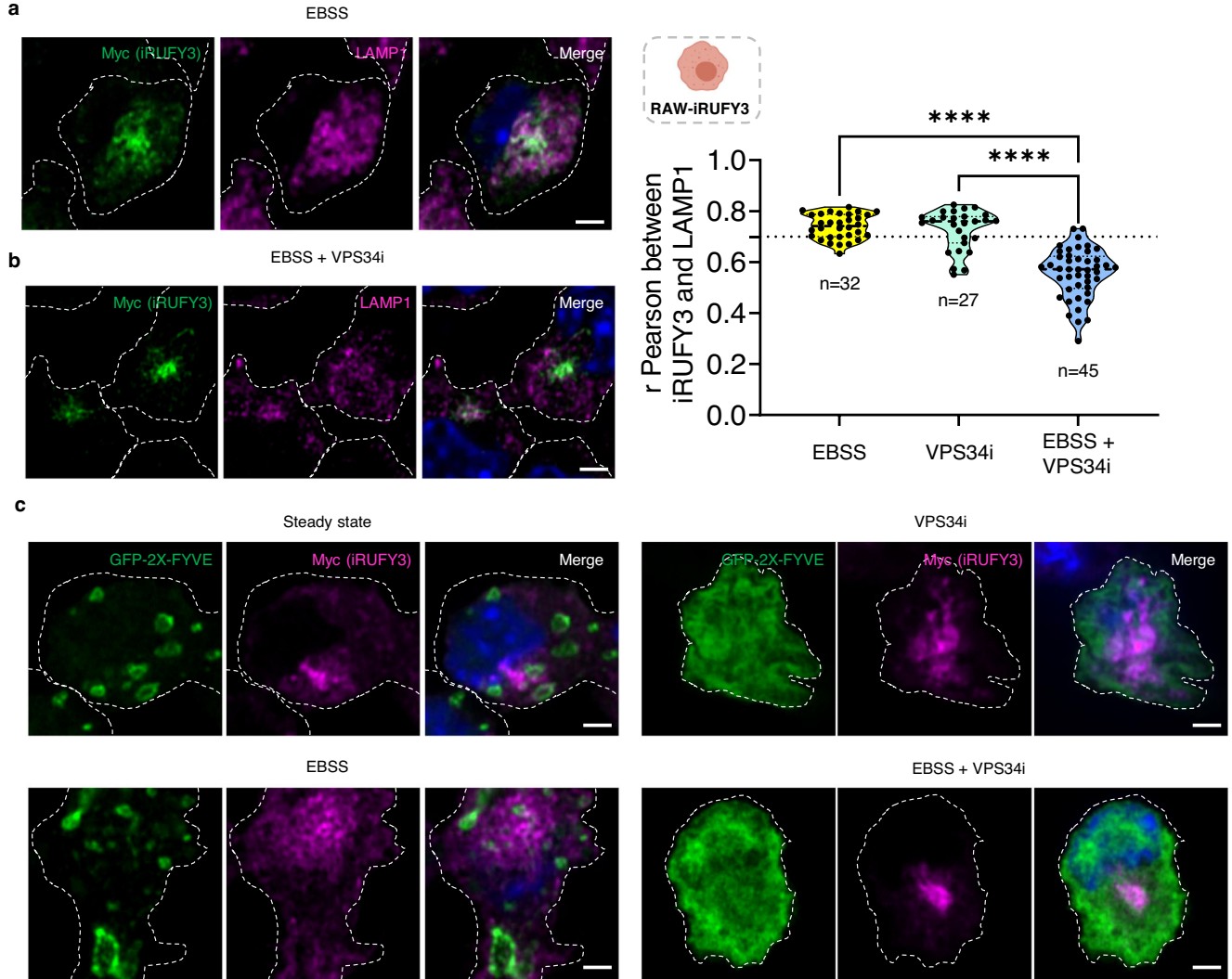

**Fig. 6 | PtdIns(3)P contributes to iRUFY3 endosomal localization. a, b** Detection of Myc-RUFY3 with EL markers LAMP1 after 6 h of starvation (**a**) or 6 h with VPS34 inhibitor (VPS34i, 5 μM). Dotted lines indicate cell boundaries. Pearson co-localization coefficients were calculated using Image J. A clear co-localization score is considered above 0.7. Numbers (*n*) of cells analyzed are indicated. Statistical significance was established using one-way ANOVA test (****$p < 0.0001$). Each dot

represents the mean of all stacks from one region of interest. **c** GFP-2X-FYVE transfection and iRUFY3 detection by AICM in RAW-iRUFY3 at steady state or nutrient starvation (EBSS) with or without VPS34 inhibition (VPS34i, 5 μM for 6 h). Scale bar 2 μm. These results are representatives of *n* = 2 independent experiments with >100 cells observed by experiments.

numbers of the proteins are as follows: Human EEA1 (NP_003557.3), Mouse EEA1 (NP_001001932.1), Human RUFY1 (NP_079434.3), Mouse RUFY1 (NP_766145.1), Human RUFY2 (NP_060457.4), Mouse RUFY2 (NP_081701.2), Human RUFY3 (NP_055776. 1), mouse RUFY3 (NP_081806.1) human RUFY3XL (NP_001032519.1), mouse RUFY3XL (NP_001276703.1), human RUFY4 (NP_940885.2), mouse RUFY4 (NP_001164112.1), human FYCO1 (NP_078789.2), mouse FYCO1 (NP_001103723.2).

## Cell culture

A complete reagents list can be found in Supplementary data. Bone marrow-derived DC were cultured with GM-CSF as described[55]. Alveolar Macrophages were obtained from 6 to 12 weeks-old mice as described[29]. Cells were resuspended in RPMI (Gibco, Invitrogen), 10% FCS, 1% pen/strep, 1% pyruvate,1% glutamine, supplemented with 2.5% GM-CSF. RAW264.7 cells were cultured as described[69]. HeLa cells were maintained in DMEM (Gibco Invitrogen)

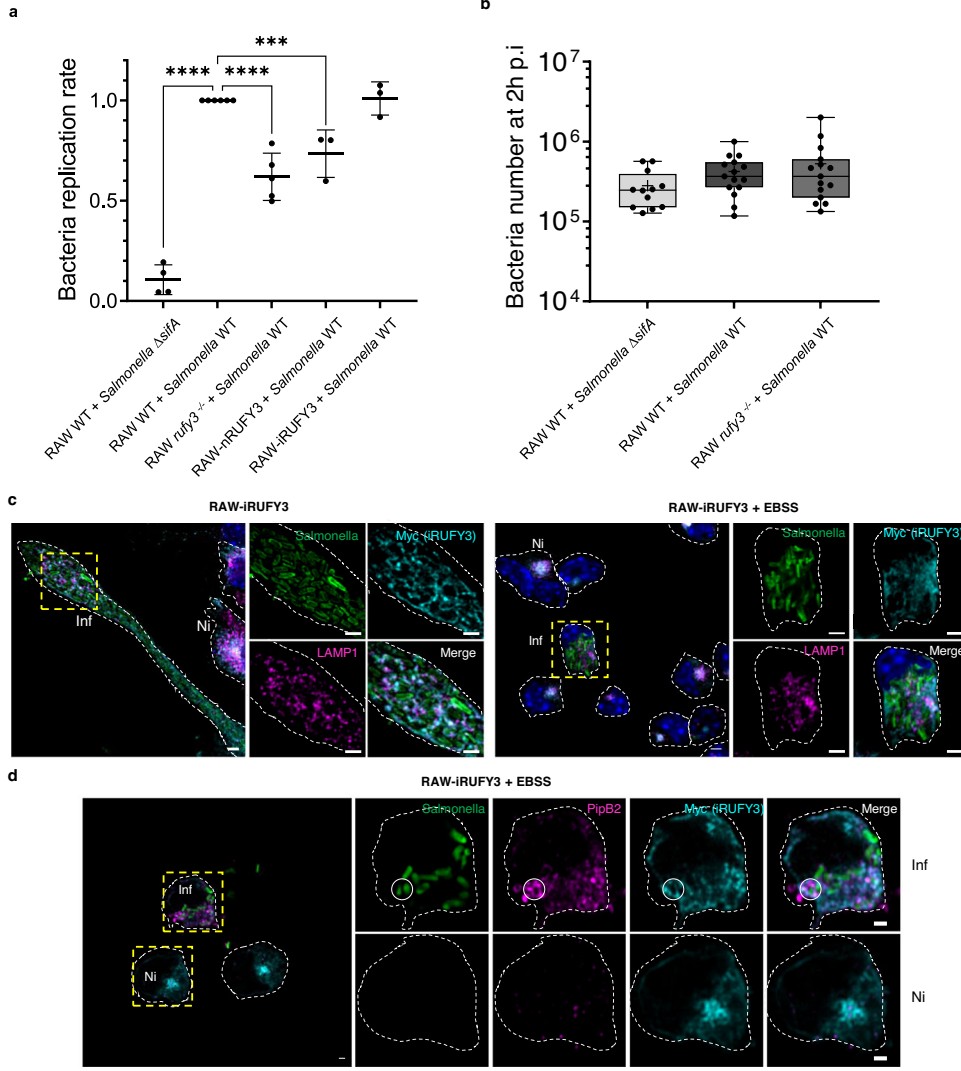

**Fig. 7 | RUFY3 is required for intracellular *Salmonella* replication in macrophages. a** *Salmonella enterica* replication was monitored by Colony Forming Unit (CFU) assay in RAW WT, RAW *rufy3⁻/⁻*, RAW-iRUFY3 and RAW-nRUFY3 cells. After gentamicin treatment, infected cells were lysed 2 h or 16 h post-infection (p.i) and colonies counted on LB agar plates. Each dot represents one independent experiment. One independent experiment represents the mean of three technical replicates (three independents agar plate). Data are presented as mean values +/- SD. Statistical relevance was established using one-way ANOVA with Holm-Šídák multiple comparisons test (*$p < 0.05$; ***$p < 0,001$; ****$p < 0,0001$). **b** Number of bacteria at 2 h p.i indicates no difference in the phagocytic rate among the different cell lines tested. Each dot represents one infection and the count of bacteria in one agar plate. For RAW WT + Salmonella ΔsifA $n = 12$ and $n = 15$ for other conditions. The boxplot data represent medians, interquartile ranges and spikes to upper and lower adjacent values. **c, d** *S. enterica* was imaged by AICM together with iRUFY3, LAMP1 (**c**) or PipB2 (**d**) either at 16 h p.i (left) or 16 h p.i with 6 h of EBSS starvation (right). Dotted lines indicate cell boundaries. Scale bar 2 μm. These results are representatives of $n = 5$ independent experiments with >100 cells observed per experiments. Infected cells (Inf) and non-Infected (Ni) are indicated in the same panels.

supplemented with 10 % FCS, at 37 °C and 5% CO2. MuTuDC1 were maintained in IMDM medium (Gibco, Invitrogen) with 8% FCS, 10 mM Hepes, 50 μM β-mercaptoethanol. DO11 T cell were grown in RPMI (Gibco, Invitrogen) with 10% FCS, 1% pyruvate, 1% HEPES, 1% non-essential amino acids. For all conditions, MAMPs and cytokines stimulation were performed with: LPS at 100 ng/mL, p(I:C) at 10 μg/mL, CpG-α ODN at 200 nM, IFN-α at 1000 U/mL and IFN-γ at 50 ng/mL. All cell lines tested negative for mycoplasma contamination using MycoAlert Mycoplasma Detection Kit (LT07-418, Lonza).

**Rufy3 gene deletion in Raw264.7 cell lines and complementation**
To generate gene-specific deletion via the CRISPR/Cas9 system, two sgRNAs targeting Exon 2 of the mouse *rufy3* gene (exon ID: ENSMUSE00000222679) were designed and cloned into CRISPR-expressing pX458-DsRed2 and pX458-ECFP[70]. The sequences of the specific sgRNAs were as follows (the protospacer adjacent motifs, or PAM for short, are underlined): TCGTTAGCCATGAGATAATT GGG and

CACCTTTCAAGCCGTGTTTC AGG. RAW264.7 cells possessing large DNA fragment deletion in *rufy3* genomic locus were obtained with the aid of fluorescent reporters coupled with the single-cell FACS sorting[70]. Primers are listed in Supplementary Table 1. Complemented *irufy3* and *nrufy3* cells were obtained by using RAW *rufy3* -/- cells transfected (jetPRIME®, PolyPlus) with plasmids coding for *irufy3* (OriGen, CAT#: MR230833; RAW-iRUFY3) or *nrufy3* transcript variants (REF CAT#: MR207512; RAW-nRUFY3) under geneticin selection. Clones were selected after western blot analysis for normal expression levels of RUFY3.

**Silencing of *arl*8b**
RAW-iRUFY3 were nucleofected (Cell Line Nucleofector™ Kit V, Amaxa Biosystems) with 100 nM siRNA against *Arl8b* (SMART pool of 4 sequences, Horizon discovery, L-056525-01-0005) and Scramble non-targeting sequences (Horizon discovery, D-001810-01-20), prior seeding on coverglasses. Treatment started 24 h after nucleofection.

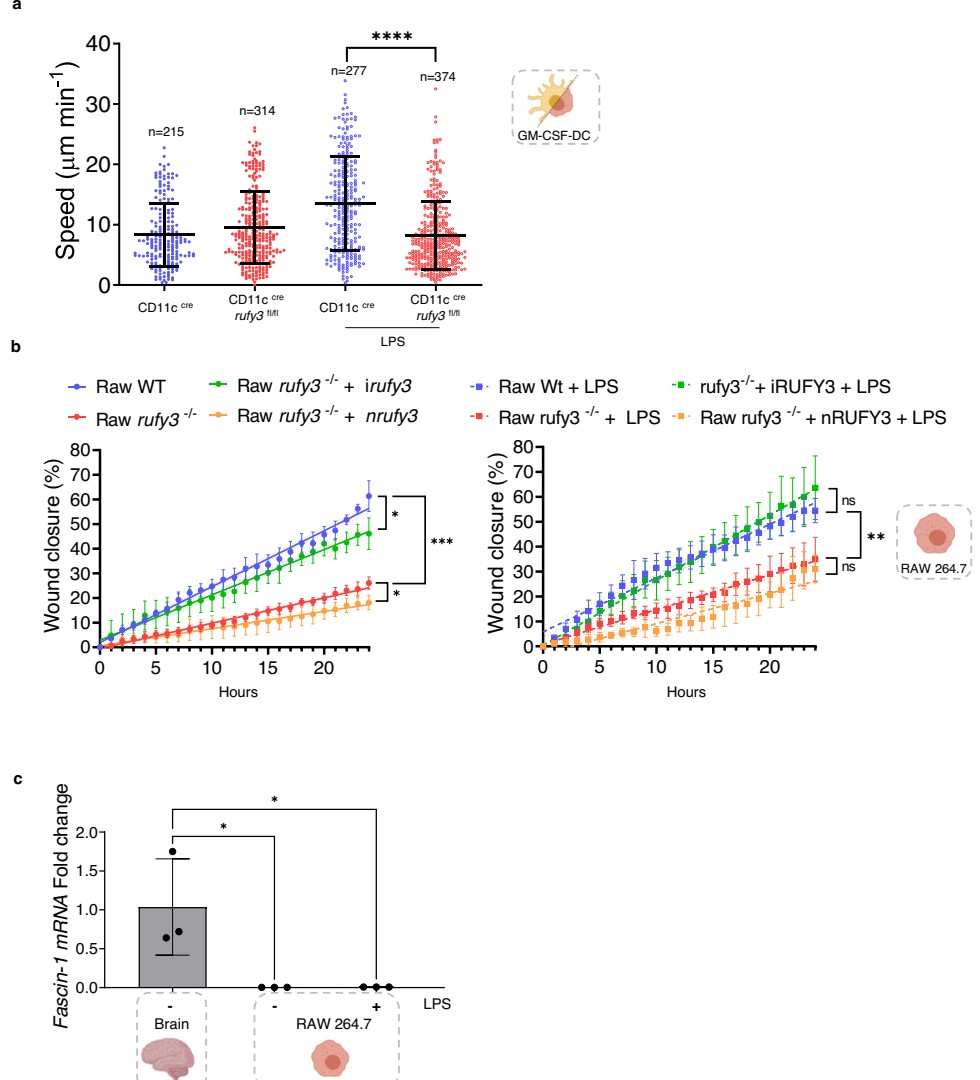

**Fig. 8 | iRUFY3 regulates migration of macrophages and DCs. a** Velocity assay for WT and CD11c^cre rufy3^fl/fl GM-CSF-bmDCs. Data represents instantaneous mean velocities with SD of migration in 4 by 5 μm fibronectin-coated microchannels of at least 100 cells per condition. *N* = 2 independent experiments were performed and each dot represents the average speed of one cell. Statistical relevance was established using unpaired Man-Whitney test (****$p < 0.0001$). **b** Wound healing assay on WT, RAW *rufy3*^-/-, RAW-iRUFY3- and RAW -nRUFY3 at steady state (left) or over 24 h LPS stimulation (100 ng/mL). For each condition, simple linear regression was performed with 99% of confidence bands of the best-fit line. Statistical relevance was established using unpaired multiple *t*-test (*$p < 0.05$; **$p < 0.01$; ***$p < 0.001$). *N* = 3 independent experiments were performed, each one in triplicate. **c** RT-qPCR quantification of *Fascin-1* from mouse brain extract and RAW264.7 macrophages stimulated or not with LPS for 16 h. *N* = 3 independent experiments were performed. Data are presented as mean values +/− SD. Statistical relevance was established using one-way ANOVA with Tukey's multiple comparisons test (*$p < 0.05$).

## Quantitative PCR

Total mRNA from cells or tissues was purified using the RNeasy Mini Kit (Qiagen). 500 ng to 1 μg of total RNA were subjected to reverse transcription using SuperScript II. Each gene transcript was quantified by SYBR Green method with 7500Fast (Applied Biosystems). The relative amount of each transcript was determined by normalizing to internal housekeeping gene expression (*gapdh*). A complete list of primers can be found in the Supplementary Table 1.

## Immunodetection

All antibodies are listed in Supplementary Data. For immunoblotting of tissue extracts, organs were taken from euthanized mice and put in 3 ml RPMI supplied with 5% FCS and 1% gentamycin. Tissues were dissociated with gentleMACS™ Octo Dissociator (Miltenyi), followed by liberase (5 mg/ml) and DNase I (150 μg/ml) digestion for 30 min at 37 °C. Cell suspensions were centrifuged and pellets were lysed in 1% triton 100X, 50 mM Tris pH7,4, 150 mM NaCl, 5 mM MgCl₂,

complemented with protease inhibitors cocktail (Roche). Lysates were centrifuged (16.000 g, 30 min, 4 °C), prior running 20-30 μg of soluble proteins in sample buffer on 3-15% gradient or 12% SDS-PAGE. Transfer was performed on PVDF membranes, which were incubated in blocking solution (TBS1X + BSA 5%) prior antibody binding and chemiluminescence detection (Pierce). For histology, lungs were taken from euthanized mice, flush with PBS and put in 10% formol, before inclusion in paraffin resin. 5 μm sections were cut prior eosin and hematoxylin or anti-CD3/B220 staining and imaging.

For Flow cytometry analysis, cells were harvested and put in a 96 v well plates, washed two times with FACS buffer (PBS, FCS 2%, EDTA 2 mM). Then, viability and surface staining were done at 4 °C during 30 min. Two more washes were performed prior fixation with 1% PFA in FACS buffer. Intracellular staining was performed using BD Cytofix/Cytoperm kit for 10 min at RT using Permwash buffer (BD, 554723). Data acquisition was performed on Canto II, LSR II UV or LSR Fortessa Symphony BD cytometers using Diva 8.0.1 and FlowJo_v10 for analysis.

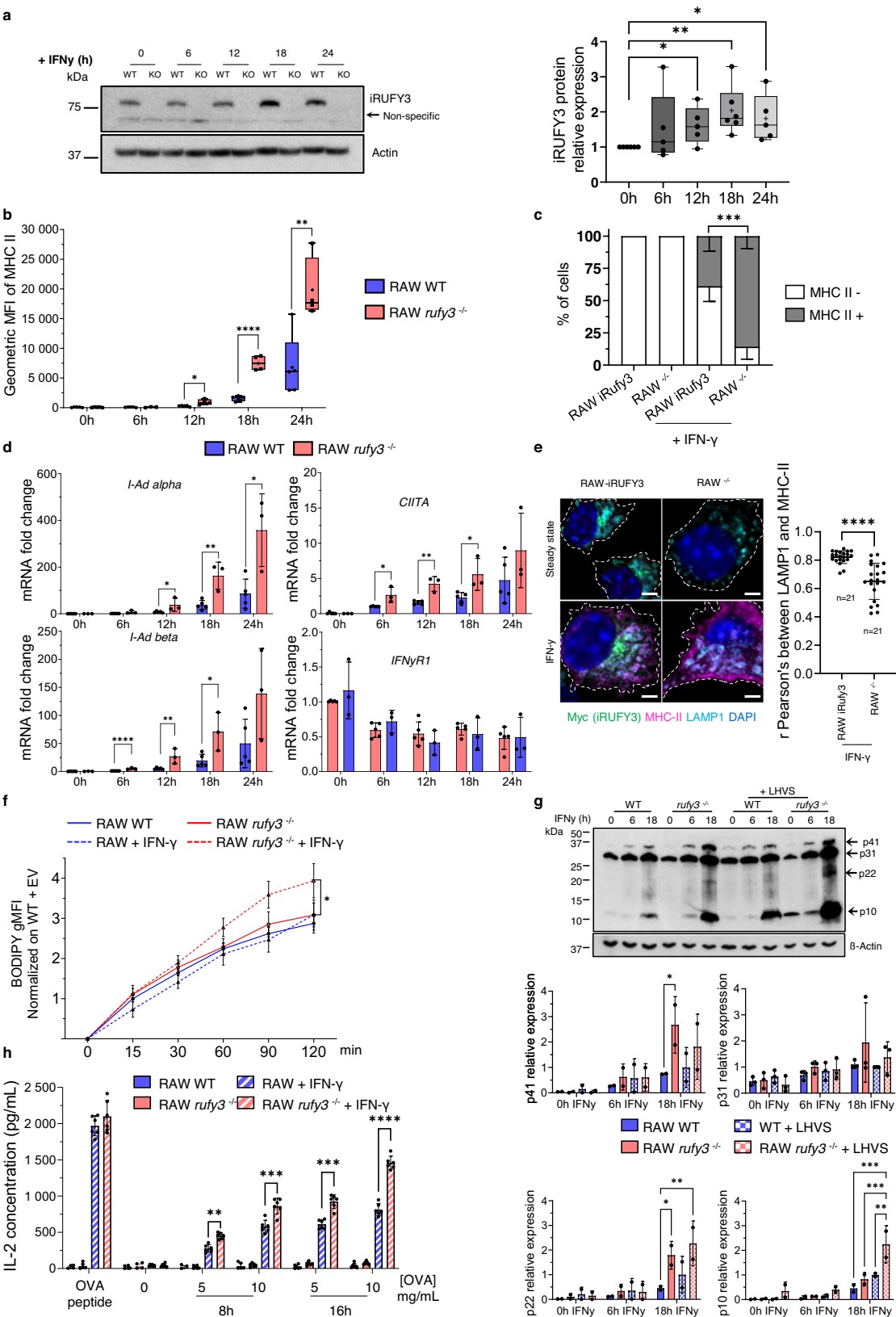

Immunoprofiling of CD11c $^{Cre}$-*rufy3* $^{loxp/loxp}$ after LPS intraperitoneal injection was performed on primary splenocytes. Spleens were harvested and cells extracted using gentleMACS™ Octo Dissociator (Miltenyi) and enzyme digestion. Surface markers profiling was performed by flow cytometry as described above, list of markers and antibodies used for cell identification and phenotyping are described in

Supplementary Table 2. All antibodies and reagents were used according to CIPHE instructions.

**Immunofluorescence confocal microscopy and image analysis**
For immunofluorescence confocal microscopy, cells were seeded on coverslips, fixed with 3.3% PFA and permeabilized 5 min with 0.1%

**Fig. 9 | iRufy3 deletion alters antigen processing and presentation in APCs.**
**a** Immunoblot quantification of RUFY3 protein levels in RAW cells stimulated with IFN-γ (50 ng/mL) during 24 h. N = 5 independent experiments. The boxplot data represent medians, interquartile ranges and spikes to upper and lower adjacent values. **b** Surface MHC-II quantification by flow cytometry after IFN-γ exposure for indicated time (50 ng/mL) in RAW and RAW *rufy3* -/-. N = 5 independent experiments for all time point in WT condition. For RAW rufy3-/- condition, n = 5 for 0 h, n = 3 for 6 h and n = 4 for 12 h, 18 h and 24 h. The boxplot data represent medians, inter-quartile ranges and spikes to upper and lower adjacent values. **c** Percentage of cells expressing total detectable MHC-II signal (MHC-II +) by microscopy after 24 h of IFN-γ stimulation. Data are presented as mean values +/- SD. **d** RT-qPCR quantifi-cation of *MHC-II* α and β chains, *CIITA* and *IFN-γ receptor-1* transcript from RAW and RAW *rufy3* -/- stimulated with IFN-γ for 24 h. Each dot is one independent experi-ment. **e** Distribution of iRUFY3, MHC-II and LAMP1 by AICM in RAW *rufy3* -/- (right)

and RAW-iRUFY3 (left) with or without IFN-γ stimulation. Scale bar 2 μm. Pearson co-localization coefficients were calculated using Image J. **f** Monitoring of OVA-DQ endosomal degradation using a fluorescence dequenching assay in response to IFN-γ stimulation. N = 4 with two duplicates. Data are presented as mean values +/- SEM. **g** RAW and RAW *rufy3* -/- stimulated with IFN-γ for indicated times (h) were treated with the cysteine protease inhibitor LHVS (1 μM) for 6 h prior immunoblot detection of Ii chain (p41/p31) and associated proteolytic fragments (p10 and p22). N = 3 independent experiments. **h** ELISA dosage of Interleukin-2 released after ovalbumin antigen processing and presentation to DO.11.10 T cells by RAW *rufy3* -/- cells and RAW after IFN-γ stimulation. N = 3 independent experiment with two technical replicates. Statistical relevance was calculated using in: **a** and **e**, unpaired *t*-test, **b** and **d**, multiple unpaired *t*-test, and **c**, **f** and **g**, two-way ANOVA with Tukey's multiple comparisons test. For all panels: *p < 0,05; **p < 0,01; ***p < 0,001; ****p < 0.0001. WT means Wild Type.

---

Triton X-100 or 0.05% Saponin. Before staining, samples were incu-bated with blocking buffer (PBS 1X, 5% FCS, 1% Glycine). Antibodies was added on samples in a wet chamber for 1 h at RT or overnight at 4 °C. Coverslips were washed in PBS three times before secondary staining. Samples were then washed in PBS and pure water prior glass mounting in ProLong™ Glass Antifade Mountant with nucleic stain (Invitrogen P36980). For Proximity Ligation Assay (PLA), cells were fixed in 3,3% paraformaldehyde and permeabilized in 0,1% Triton-X100. Duolink PLA rabbbit/mouse was used, according to manufacturer instructions (Merck, DUO92101).

Images were captured with a Zeiss LSM880 and Zeiss LSM780 confocal microscope using a 63x/1.40NA M27 Plan Apochromat oil objective. High resolution imaging was performed using the Airyscan module in Zeiss Black v3.9 (Carl Zeiss AG, Jena, Germany). Samples were excited at 405, 488, 561 and 633 nm individually. Dicroïc mirror MBS-405 + MBS 488/561/633 was used for laser. Emission filter com-binations used was BP 420-480 + LP 605 / BP 495-550 / BP 570-620 + LP 645 for 405, 488, 561, 647 nm laser respectively. Supplemental dicroïc mirror were used for 561 and 647 nm laser with SBS SP 615 and SBS LP 660.

Zen Blue 3.5 software was used to process the acquired images. The Airyscan filtering (Wiener filter associated with deconvolution) was set to the default filter setting of 6.1 in 2D, prior deconvolution and pixel reassignment to improve SNR. Levels of co-localization (r Pear-son and Mander's Overlap Coefficient) and cytofluorograms were quantified using JACoP plugin[71] and manual definition of the Regions of Interest (ROI). Final r Pearson coefficient and MOC values corresponds to the mean of all stacks of the ROI after subtracting the threshold. All images were acquired in Z-stack mode prior assembly and recon-stitution in IMARIS 9.9 software to obtain 3D rendering. For some images, drift was corrected by applying images stabilizer plugin from Image J software. For each 3D images, frame was put and X, Y, Z axis were represented at the top left corner. For each image, surfaces were added on nucleus. Cell shapes were manually added as a white dotted line for all images. For PLA, spots quantification indicating a proximity of targets <40 nm. Spots were counted and distance to the nearest neighbor measured. Voxel gating was extracted from IMARIS 9.9 with coloc plugin and mask from colocalization region were obtain after subtracting the threshold.

### Clustering index
To obtain the Clustering index, a centroid based method was used with dedicated macro on ImageJ (https://github.com/Imagimm-CIML/Determining-cell-polarisation, Macro_polarisation_polar_dispersion.ijm). Masks on LAMP1, ARL8b and nucleus staining were applied indi-vidually. The coordinates of the centroid and the distance between LAMP1/nucleus or ARL8b/nucleus centroid were calculated and nor-malized on cell radius (calculated from Feret diameter) to obtain a score between 0 to 1 (Clustering index). Polarized cells are defined with a unique perinuclear site and a score close to 0.5. Conversely, non-

polarized cells with limited accumulation at a unique site will have a shorter centroid between nucleus and organelle mask with a score close to 0.

### 360-ASOD method
360-ASOD (360° Angular Scanning monitoring Organelle Distribution) has been filed on Github (Imagimm-CIML/360-Angular-Scanning-monitoring-Organelle-Distribution (github.com)). A mask on nucleus was generated to identify the cell centroid. From this centroid, an image with polar coordinates is created from the ImageJ plugin "Polar transformer". From this radial image, a dedicated macro (available in the github link) was used to extract the angular profile of the orga-nelles signal summing the signal in the radial direction. This profile is generated by counting the number of pixels in the radial direction of the nucleus centroid above an intensity threshold for each angle (from 0 to 360 degrees). Each profile was then normalized on the sum of the signal to transform the angular profile in probability. Moreover, the profile was centered to 0 between −180 to 180 degrees on the max-imum intensity after applying lowess regression to smooth the signal. The variance was then extracted for each profile and plotted. Finally, the mean profile for each condition was also generated. A polarized cell will have a lower variance, meaning a concentration of organelle over a small range of angles around 0. Conversely, a non-polarized cell will have a distribution surrounding the nucleus with a high angle range (ideally a flat curve). Representation of this method is shown in Supplementary Fig. 2b.

### Cell fractionation
For each condition, $15.10^6$ cells were treated in hypotonic solution (10 mM triethanolamine, 10 mM acetic acid, 1 mM EDTA, 250 mM sucrose, titrated to pH 7.4) for 15 min at 4 °C. Cells were lysed by 8 passages in a ball bearing cell cracker at at 4 °C, prior centrifugation to collect post-nuclear supernatants (PNS). PNS were then centrifuged at 16,000 g for 30 min to obtain the membrane-enriched protein (Mb) and cytosolic protein (Cy) fractions.

### Cell migration and velocity
Scratch wound healing assays were performed with cells seeded to confluence in lab-tek 2 chambers (Thermo Scientific, Ref 155380). Prior scratching, cells were treated with EBSS medium for 2 h to synchronize cell cycle. Scratching was performed with 20-200 μL sterile pipette tips and wound closure was acquired every hour using a video microscope (Zeiss, Axio-observer, 10x/0.25 NA) over 24 h at 37 °C in 5% CO2. Rate of wound closure was calculated using Wound Healing Tool plugin on ImageJ from MRI. Wound closure was determined by the equation: Wound Closure % = $(\frac{A_{t=0}-A_{t=n}}{A_{t=0}}) \times 100$ where $A_{t=0}$ is the initial wound area, $A_{t=n}$ is the wound area after *n* hours of the initial scratch, both in μm². 

Velocity of CD11c^Cre and CD11c ^Cre-*rufy3* ^loxp/loxp GM-CSF BMDCs was performed as previously[72]. Briefly, DCs were taken at day 10 of culture

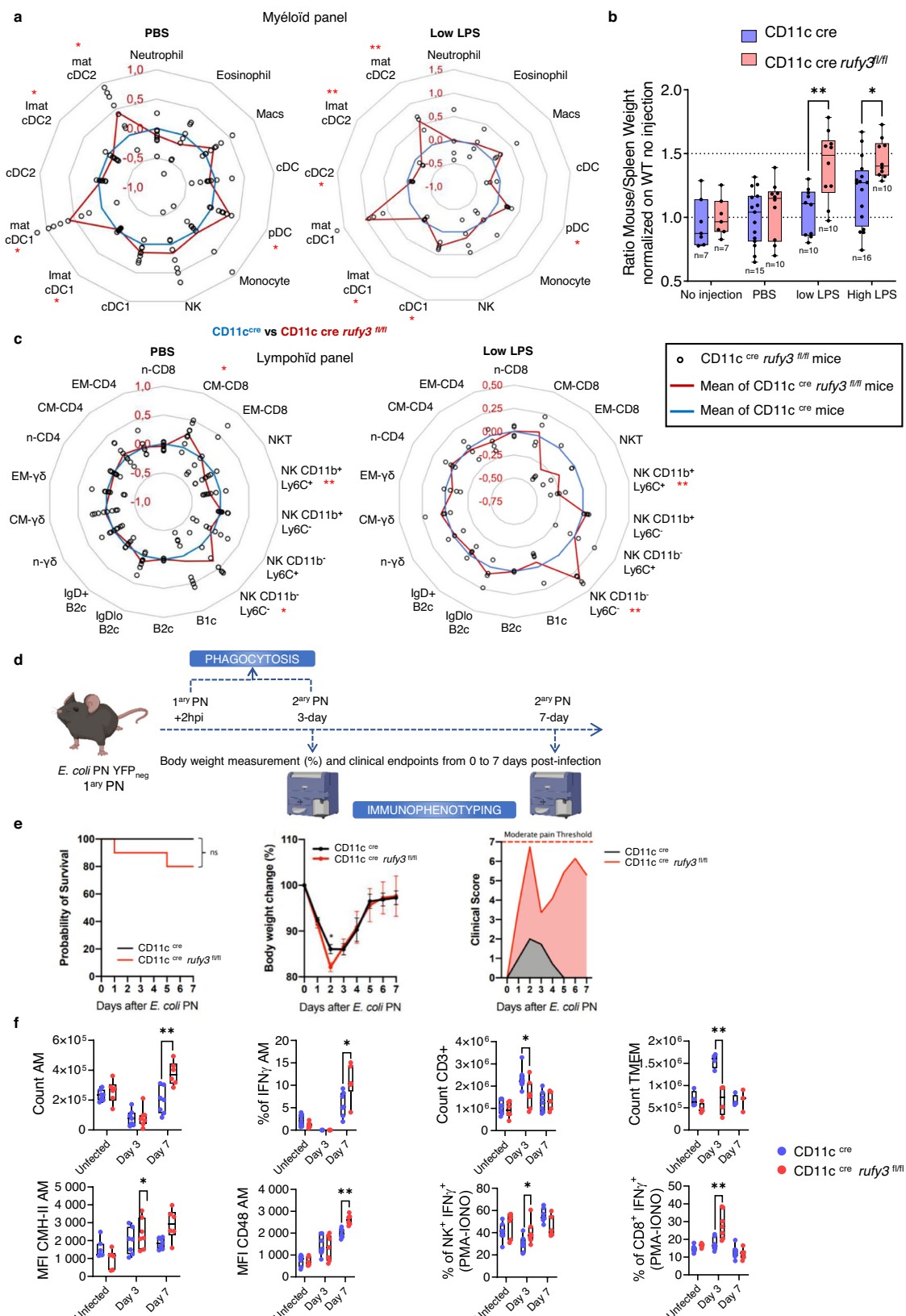

and put (iDCs) in microfluidic channels (5 × 5 μm) or challenged with a 30 min pulse of 10 ng/mL LPS (LPS-DCs) prior washing and channel introduction. Hoestch dye was added to stain nuclei and follow cells. Images were acquired every 2 min for over 16 h at 37 °C in 5% CO2, with a 20X objective video microscope (Zeiss, Axio-observer). Data are shown as the median of velocity.

### In vitro Salmonella infection

*S. enterica* infection was performed as described[42]. Briefly, Control, complemented cells and RAW rufy3$^{-/-}$ cells were seeded in six well plates with 1.10$^6$ cells per well. *S. enterica* strains were treated 30 min on ice with mouse serum and added at MOI 20. After phagocytosis, gentamicin was added at 100 μg/ml during 1 h and 5 μg/

**Fig. 10 | Rufy3 deletion in the CD11C+ cell compartment is pro-inflammatory.** **a** and **c** Immunophenotyping of myeloid (**a**) and lymphoid (**c**) splenocyte populations from CD11c$^{cre}$ and CD11c$^{cre}$ *rufy3*$^{fl/fl}$ mice by flow cytometry after IP injection of PBS (left) or LPS (1,5 ng/mg) (right), shows signs of mild spleen inflammation. Red line represents the average cell numbers in the spleens of CD11c$^{cre}$ *rufy3*$^{fl/fl}$ relative to control animals (blue line). For PBS condition, 7 CD11c$^{cre}$ mice and 8 CD11c$^{cre}$ *rufy3*$^{fl/fl}$ were used. For low LPS, 3 CD11c$^{cre}$ mice and 4 CD11c$^{cre}$ *rufy3*$^{fl/fl}$ were used. $N = 4$ for PBS injection and $N = 3$ for LPS injection. **b** Splenomegaly was revealed by monitoring spleens weight from control and CD11c$^{cre}$ *rufy3*$^{fl/fl}$ animals after LPS IP injection (1,5 ng/g or 10 μg). N was added directly on graph. Each dot represents the ratio between the weight of the spleen and the total weight of the corresponding mouse. Statistical relevance was established using two-way ANOVA with Šídák's multiple comparisons test (*$p < 0,05$; **$p < 0,01$). The boxplot data represent medians, interquartile ranges and spikes to upper and lower adjacent values. **d** Graphical abstract of the acute pneumonia model. **e** Disease scores of WT and CD11c$^{cre}$ *rufy3*$^{fl/fl}$ mice upon *E.coli* inhalation and primary pneumonia ($N = 10$) recovery over 7 days. Kaplan-Meyer survival curve (left), daily weight monitoring (middle) and overall pain score (right) graphs show increased severity of the disease in infected CD11c$^{cre}$ *rufy3*$^{fl/fl}$ animals. $N = 10$. **f** Flow cytometry profiling of total lung immune cells indicates increased counts in IFN-γ-producing resident macrophages, CD8 + T and NK cells in infected CD11c$^{cre}$ *rufy3*$^{fl/fl}$ compare to control animals. $N = 7$ for CD11c$^{cre}$ mice and $n = 6$ for CD11c$^{cre}$ *rufy3*$^{fl/fl}$ mice. The boxplot data represent medians, interquartile ranges and spikes to upper and lower adjacent values. Statistical relevance was established by multiple unpaired Mann–Withney test (*$p < 0,05$; **$p < 0,01$).

ml for the rest of the experiment at 37 °C. Infected cells were lysed 2 h and 16 h after infection by adding 0,1% triton to harvest bacteria. The lysate was diluted in cascade fire times with a factor of 10 prior plating on agarose. The drop was dried at room temperature and the number of colonies were counted for each dilution factor after 24 h. The final replication factor was expressed as ratio of colonies numbers at 16 h p.i. on the numbers at 2 h p.i.

### Antigen processing and presentation assay
Ovalbumin-DQ degradation was followed using $1.10^5$ RAW264.7 treated or not with IFN-γ for 18 h. Ovalbumin conjugate with BODIPY was added at 100 μg/ml. Negative control was kept at 4 °C on ice during OVA-BODIPY internalization to determine background fluorescence. After ovalbumin processing and BODIPY dequenching, cells were washed with PBS and processed for flow cytometry. Final BODIPY-MFI was calculated using $(Xn_{37°C} - X0_{37°C}) - (Xn_{4°C} - X0_{4°C})$ where "Xn", "X0" are conditions $X$ after $n$ min of ovalbumin processing at 37 °C or 4 °C. Antigen presentation was performed using specific DO11.10 T cells. $3.10^4$ RAW 264.7 (RAW) cells were put in 96 V-wells plate after 20 h IFN-γ stimulation or not. After PBS, washes, ovalbumin was added at 5 mg/mL or 10 mg/mL for 8 h to 16 h. As positive control, OVA peptide 323-339 alone was added to the cells at 100 μg/mL for 8 h. RAWs were washed with PBS and DO11.10 T cells added at 5:1 ratio ($1,5.10^5$ T cells) for culture overnight at 37 °C and 5% CO2. Culture supernatants were harvested and IL-2 concentration was measured by ELISA (Invitrogen, # BMS601) and colorimetric reading at 450 nm.

### Statistics
All statistics were done using Prism 9 software. The most appropriate statistical test was chosen according to each set of data, as indicated in figure legends with $p$-values *$p < 0.05$; **$p < 0.01$; ***$p < 0.001$; ****$p < 0.0001$.

### Reporting summary
Further information on research design is available in the Nature Portfolio Reporting Summary linked to this article.

## Data availability
All relevant data supporting this study are presented in the manuscript and supplementary information. Raw data and uncropped western blots are available in the Source Data file data are provided with this paper (https://doi.org/10.6084/m9.figshare.22725707). Gene description and sequences are available from NCBI (https://www.ncbi.nlm.nih.gov). Gene expression profiles from immune cells are available from the Immunological Genome project. https://www.immgen.org. Microscopy data are available on reasonable request to the corresponding author. Source data are provided with this paper.

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

## Acknowledgements

R.C. received a fellowship from the Fondation de la Recherche Médicale" (FRM). The P.P. laboratory is Equipe FRM sponsored by the grant DEQ20180339212, and grants from the Institut National du Cancer (INCA) (PLBIO17-187), Fondation ARC "PGA 2021-2025-CHARP" and Agence Nationale de la Recherche (ANR) AAPG2021-STIM. This work was also supported by the project PTDC/BIA-CEL/28791/2017 and POCI-01-0145-FEDER-028791, as well as 2022.03217.PTDC, UIDB/04501/2020 and UIDP/04501/2020, funded by FEDER, through COMPETE2020—Programa Operacional Competitividade e Internacionalização (POCI), and by national funds (OE), through FCT/MCTES. B.S. acknowledge support from National Key R&D Program of China (2021YFA1301400), the National Natural Science Foundation of China (31930035, 32061143028), Shanghai Science and Technology Commission (20410714000, 22JC1402600). We thank the "Shanghai 1000 talents" program and the Maratona da Saúde for their support. A.R. received grant from l'Agence Nationale de la Recherche (ANR) «JCJC-PROGRAM». We acknowledge the PICSL imaging facility of the CIML (ImagImm), member of the national infrastructure France-BioImaging supported by the French National Research Agency (ANR-10-INBS-04). We thank Lionel Spinelli at CIML for statistics and mathematical assistance. Illustrations in the different figures were created with BioRender.com. We thank Sarah Pamukcu for helping with illustrations design.

## Author contributions

P.P., E.G., B.N. and Ré.C. conceived the project and analyzed the data. Ré.C designed and conducted most of the experiments. Z.L., B.S. and Y.L. realized and contributed cell lines generation. Ré.C., Ra.C., M.F. contributed to cell imaging and data quantification. C.S., E.S. and V.C. contributed to mouse experiments and biochemistry. R.R. provided meningeal extracts. M.-G.D and A.-M.L.-D. performed velocity assay. C.J., M.D. and A.R. contributed to the infection mice model and analysis. S.M. and Ré.C. contributed to Salmonella infection experiments. L.C. helped to histology section. P.P. wrote original draft and P.P., E.G., Ré.C., B.N. wrote the manuscript with input from all the authors. P.P., E.G. and C.A. contributed to get funding to perform the study.

## Competing interests

The authors declare no competing interests.
