## [Peer Review File · Nature Communications]

RUFY3 regulates endolysosomes perinuclear positioning, antigen presentation and migration in activated phagocytesREVIEWER COMMENTS

Reviewer #1 (Remarks to the Author):

In this manuscript, Char et al. highlight the function of the FYVE domain-containing longer isoform (iRUFY3) of mouse *rufy3*, which is preferentially expressed in immune cells, in regulating the positioning of endolysosomes (ELs) in the perinuclear cloud/region of the cells as well as in regulating cell migration, intracellular *Salmonella* replication, and antigen presentation. Here, the authors first provide evidence that iRUFY3 is specifically expressed in bone marrow, lymphoid organs, and immune cells (including mouse macrophage cell line-RAW264.7), while the canonical shorter *rufy3* isoform (nRUFY3) is mostly expressed in the brain. They show that the expression of iRUFY3 is upregulated upon treatment with different MAMPs, IFN α , as well as upon nutrient starvation. Using reconstitution of CRISPR edited RAW cells lacking RUFY3 expression (RAW RUFY3 KO cells) with tagged-forms of iRUFY3 and nRUFY3, iRUFY3 was shown to localize to LAMP1-positive ELs, while nRUFY3 displayed mainly cytosolic distribution. The authors also showed that iRUFY3 localization to LAMP1/lysosomal GTPase Arl8b-positive ELs was increased upon LPS stimulation (and also under the starvation condition) and that iRUFY3 promoted the perinuclear clustering of these cellular compartments. The authors concluded that RUFY3 is required for maintaining the perinuclear pool of ELs through its interaction with Arl8b and promotes retrograde transport of ELs. The authors then focus on the role of RUFY3 in regulating the replication of intracellular pathogens, using *Salmonella* as a model organism. As previously reported, perinuclear positioning of *Salmonella* containing vacuoles (SCVs) is important for the development of replicative niches. In the absence of iRUFY3, *Salmonella* replication was less in RAW macrophages. To elucidate the function of RUFY3 in vivo, the authors generated a transgenic mouse model lacking expression of RUFY3 in CD11c-expressing cells, including most DC subsets and alveolar macrophages. Interestingly, the *rufy3*-deficient GM-CSF-DCs (as well as RAW macrophages) were less migratory compared to wild-type cells when challenged with LPS, and this was not due to alteration in the actin cytoskeleton. The authors further characterized the CD11c+ cell compartment in a *rufy3*-deficient mouse model and found it to exhibit a more pro-inflammatory phenotype. This was also in line with higher levels of MHC class II molecules on the surface and better presentation of exogenous antigens.

The manuscript is filled with a ton of data on iRUFY3 function, and with a few exceptions described below, the data is largely convincing and well done. The antigen presentation experiment showing that iRUFY3 depletion interferes with MHC class II surface levels and exogenous antigen presentation in Fig. 5 is interesting and needs further mechanistic insights for a solid conclusion regarding its physiological significance.

While the data presented is interesting, it is mostly descriptive of different phenotypes observed upon iRUFY3 depletion in macrophages. For instance, the authors have made very interesting observations that RUFY3 regulates macrophage cell migration and exogenous antigen presentation. However, the authors have not explored the mechanistic basis for these phenotypes. The manuscript in its present form is incomplete, and the following points warrant attention prior to publication:

Major points:

1. In the text on page 4, the authors describe the main difference between the nRUFY3 and iRUFY3 variants of the *rufy3* gene as the presence of an extended c-terminal region of 150 amino acids containing the FYVE domain. This is a bit confusing. As the difference between the variants is actually 200 amino acids and not 150 amino acids, that is what is shown in Fig. 1a. Are the authors sure that the nRUFY3 referred to here is NP_001346137.1 and not NP_001276704.1, which is 519 a.a. long? This information needs to be clarified and accordingly changed in the text.
2. There is variation in the size (kDa) of the iRUFY3 band in several western blots, for example, Figs. 1c/d to 1e/h/i and extended Fig. 1d and 1e. Can the authors clarify the reason for this variation in mobility of the iRUFY3 protein band? Is it possible that it is due to differences in different PTMs in different cell types? This needs to be clarified.
3. In Fig. 1c and the extended Fig. 6d, why is the b-actin signal missing in lanes in which HeLa cell lysate expressing nRUFY3 and iRUFY3 is loaded? This is a bit surprising, and no explanation is

provided in the text or figure legends.

4. What is the rationale for doing the experiment shown in Fig. 1e? Why was LPS treatment given to check the dimerization of iRUFY3? It's already reported in a recent study (Kumar G, Chawla P, Dhiman N, Chadha S, Sharma S, Sethi K, Sharma M, Tuli A. RUFY3 links Arl8b and JIP4-Dynein complex to regulate lysosome size and positioning. *Nat Commun.* 2022 Mar 21;13(1):1540. doi: 10.1038/s41467-022-29077-y. PMID: 35314681; PMCID: PMC8938454) that the CC domain of RUFY3 is responsible for dimerization. However, the schematic depiction of the dimeric form of iRUFY3 in Fig. 1e reflects the involvement of the RUN domain in dimerization, although no experimental evidence is provided to support this claim. However, the authors hypothesize in the result section about the involvement of the CC domain in the dimerization of iRUFY3. The authors should either revise the schematic or provide experimental evidence for the involvement of the RUN domain in dimerization.

5. In general, there is a lot of discrepancy between what is mentioned in the text and what is described in the figure legends. Just an example. In the text on page 5, the authors report an increase in the levels of iRUFY3 in BMDMs upon LPS treatment. However, in the legend of Fig. 1i, the cell type mentioned is RAW macrophages. Firstly, the authors should correct such a mistake. Secondly, if the experiment was performed in BMDMs and not RAW macrophages, then what happens to iRUFY3 levels in RAW macrophages upon treatment with LPS and with other MAMPs treatments?

6. The authors state and illustrate in Figs. 1h and 1i that LPS stimulation induces or augments iRUFY3 expression (or with other MAMP treatments). However, Figs. 1c and 1e show a reduction in the band intensity of iRUFY3 in response to LPS stimulation. These experiments need quantification of blots to better illustrate the claims.

7. In extended Fig. 1c, there is a ~50% drop in the iRUFY3 mRNA fold change (compare iRUFY3+nRUFY3 to iRUFY3 alone). However, as per the data provided in Fig. 1, there should be no reduction since iRUFY3 is the only form found in DCs. The authors should provide an explanation for this observation.

8. In Fig. 2 and extended Fig. 2, the author shows that iRUFY3 is recruited to pericentriolar LAMP1+ endosomes in response to LPS activation or upon nutrient deprivation. To substantiate this claim, authors should conduct biochemical experiments (such as membrane cytosol fraction or EL separation) following LPS treatment and starvation. In addition, the rationale for six hours of starvation with EBSS is unclear. A short period of starvation is not enough for the localization of iRUFY3 to LAMP1-positive compartments.

9. For the Pearson coefficient quantification graphs shown in Fig. 2b and extended Figs. 4b and 4c, the authors must describe the whole approach used for analysis in the material and methods section. Furthermore, no information is provided on the number of cells used for quantification, which should be included in all the figure legends. Some samples have a small number of data points, so the authors should increase the number of data points to be consistent.

10. To determine the increase in iRUFY3 recruitment on EL or other organelles, it is important to quantify (Mander's coefficient) what fractions of EL or other organelles colocalize with iRUFY3 upon LPS treatment and starvation in comparison to untreated cells.

11. Both starvation and LPS treatment induce EL clustering in the juxta-nuclear region, and a smudge is visible in the majority of confocal micrographs. Therefore, quantifying protein colocalization under such conditions is challenging. To get a better idea of how much more iRUFY3 is being recruited to ELs, the authors should do these experiments in the presence of microtubule polymerizing inhibitors.

12. In Fig. 3, what do the dots in the various graphs represent? There is no information provided in the figure legend or in the M&M section. To strengthen the PLA data, the authors should biochemically (co-ip) test the interaction between Arl8b and nRUFY3 and iRUFY3. Also, what happens to iRUFY3 localization in cells lacking Arl8b at steady state, during LPS stimulation, and upon starvation? Will it depict cytosolic distribution like nRUFY3 in cells lacking Arl8b? Only the FYVE domain of iRUFY3 is solely sufficient for perinuclear clustering of ELs?

13. For the experiments shown in extended Figs. 4 and 5, the rationale for including the EBSS treatment along with YM201636 or VPS34i is not clear. From the graph shown in extended Fig. 4c, it's not clear why iRUFY3 colocalization is decreased with LAMP1 under the condition of YM201636 treatment? The authors should clarify this point.

14. The experiment shown in Fig. 4a is difficult to comprehend, and the author's assertion about the increase in the quantity and volume of ELs based on this one experiment necessitates further investigation. There is no correlation between the increase in EL number and volume shown in this experiment and the confocal images and staining of LAMP1 and LAMP2 in Figs. 2 and 3. Furthermore, no information on how this experiment performed is provided in the M&M section.

15. In Fig. 4d, the authors show the recruitment of iRUFY3 around the SCV and suggest that iRUFY3 participates in the maturation of the SCV. However, no experiment has been conducted to corroborate this observation. The authors should check the positioning of SCVs in RUFY3 KO cells. Also, it should be determined if the recruitment of iRufy3 to SCVs is dependent on or independent of Arl8b. Again, the rationale for combining starvation with a Salmonella infection is not clear.

16. The role of RUFY3 in regulating cell migration is interesting, but it's not thoroughly investigated. The authors can decide to remove this data set or else should investigate in more detail by looking at the staining of focal adhesions, integrins surface levels, etc. in RUFY3 KO cells. Though the authors claim that they do not see any major change in the actin cytoskeleton in RUFY3 KO cells, the number of filopodia seems to be higher in RUFY3 KO cells based on the confocal micrographs shown in extended Fig. 7c.

17. The effect of rufy3 depletion on antigen processing and presentation by APCs is very interesting but not mechanistically explored by the authors. The authors should test if the lack of iRUFY3 effects the recycling rate of peptide-bound MHC-II since, in the absence of iRUFY3, ELs are positioned closer to the plasma membrane. Such experiments will provide better insights for iRUFY3 role in immune cells and will also be useful to comprehend the pro-inflammatory phenotype of the CD11c+ compartment observed in the mouse model of RUFY3 KO as shown in Fig. 6 and extended Figs. 8 and 9.

18. The figure legend for extended Fig. 10 needs to be described in detail.

19. There have been two recent publications on RUFY3 as an Arl8b effector and how the RUFY3-Arl8b complex regulates perinuclear lysosome positioning (PMID: 35314681 and 35314674). Though the authors have cited these publications in the manuscript, at some places one or the other work is not cited. Since the results shown in both of these studies are complementary and these studies came out at the same time, it is important to give equal credit to both research groups. Therefore, the two publications should be cited together wherever they are referred to in the manuscript text.

Minor points:

Note: There are mistakes as well as missing information in the manuscript text, figure labels, and figure legends.

1. The labeling of "RAW 264.7" cells should be kept consistent both in manuscript text, figure labels, and legends.

2. RUFY3 CRISPR-edited knockout cells should be mentioned as "RAW RUFY3-/-" instead of "RAW-/-." This information should be corrected in the manuscript text, figure labels, and figure legends.

3. The evidence of RUFY3 KO is provided as a series of immunoblots that lack detectable protein. Given that the KO cells are single-cell clones, the authors should also provide a summary of their sequencing data to show that the rufy3 gene was disrupted.

4. "Fig. 11" does not exist as mentioned in the text; it should be changed to "Fig. 1i."

5. In manuscript text, "zinc finger motifs" is mistyped as "zing finger motives." Also, "nutrient starvation" is mistyped as "nutriment starvation".
6. In Fig. 3b (the right-side graph), what the blue and white bars represent is not labeled.
7. In the extended Fig. 5d graph, the labeling should be aligned properly.
8. The figure legend for Fig. 5e is lacking information on what the dot represents.
9. The figure legends lack information regarding certain elements, such as Fig. 1c (MLN and ILN) and Fig. 1e (W and K).

Reviewer #2 (Remarks to the Author):

In this study, Char and colleagues explore the expression and functions of iRUFY3, a rufy3 splicing variant, focusing on immunity. rufy3 has 2 splicing variants: nRUFY3 (without FYVE domain) and iRUFY3 (with a FYVE domain). While nRUFY3 interacts with the actin network and has been mostly studied in the context of nervous system development, the iRUFY3 isoform has recently dragged attention in a back-to-back publication in Nat. Comm. (Kumar et al., 2022; Keren-Kaplan et al., 2022) showing that iRUFY3 interacts with Arl8b to position endosomes. Here, Char and colleagues provide more insights into the tissue-specific expression of iRUFY3. Following their observation of a preferential expression of iRUFY3 in (activated) immune cells, the authors investigate different functions of iRUFY3 in immunity. They observe that the endosome clustering in LPS-activated macrophages is iRUFY3 dependent, while the migration of macrophages and DC as well as antigen presentation in activated cells is reduced by iRUFY3. Besides, iRUFY3 contributes to Salmonella intracellular replication in macrophages, potentially by contributing to the formation of a mature/viable bacterial vacuolar niche. Finally, the deletion of rufy3 appears pro-inflammatory, suggesting an overall versatile function of iRUFY3 in modulating immune reactions.

This study explores many aspects of iRUFY3 and provides several novel and relevant insights, in particular on rufy3 tissue-dependent alternative splicing and iRufy3 roles in immune cells. However, the different directions of the paper are not always connected – at least at the experimental level. As a result, the proposed model and the title assume many links that are not (yet) supported by the data. Besides, the part of the manuscript on PI3P is not convincing and requires more in-depth investigation.

Major comments:

1_ General. As currently written, the introduction and some elements of the results undermine the previously published work showing that iRUFY3 interacts with Arl8b to regulate lysosome size and positioning (Kumar et al., 2022; Keren-Kaplan et al., 2022). Although the reference is indeed quoted, the overlap with some of the claims of the paper is not transparently exposed, which could wrongly suggest that the interaction of iRUFY with Arl8b and its role in endosome positioning is a novel finding from Char et al. I would recommend the authors to be more upfront on what has been published and what is the adding of their work. This will help the readers to focus on the key findings of this manuscript.

For instance:

- in the sentence "several effectors have been identified for mammalian ARL8, [...] and recently the RUN and FYVE domain-containing protein 3 (RUFY3)" (page 3), RUFY3 should be replaced by iRUFY3 to be clear that this is the same isoform as in this study.
- in the sentence "iRUFY3 drives the pericentriolar clustering of ARL8b/LAMP1+ ELs and is closely linked to PtdIns(3)P production." (page 4), as these elements were previously published, references should be added.

2_ iRufy3 endosomal localization and PI3P+ compartments. The authors display many microscopy panels in the main and extended figures showing the endosomal localization of iRUFY3 in immune cells upon different treatments (LPS, EBSS) (see also minor points). The authors wrote: "Although sorting and recycling endosomes (SE, RE) markers EEA1, RAB11A and Syntaxin 6 were found in close vicinity with iRUFY3, the late endosomal marker LAMP1 displayed the best co-localization with iRUFY3 (Fig. 2b)." But actually, EEA1 is not visibly in close vicinity with iRUFY3 on any of the 3 tested conditions. This is an important point as EEA1 is the textbook definition of a PI3P-bound protein, which is less expected for LAMP1.

Then the authors explore the FYVE-domain-dependent localization of iFURY3 on PI3P+ compartments. For this, they use PIKFyve and VPS34 inhibitors to change the internal pool of PI3P. The impact of PIKFyve and VPS34 in PI3P should be better explained in the text. I.e, VPS34 phosphorylate PI into PI3P and PIKFyve phosphorylate PI3P into PI3,5P2, making the inhibition of VPS34 and PIKFyve decreases or increases intracellular PI3P respectively. The result should be interpreted/discussed along that line. In particular, the localization of iFURY3 does not seem to be affected by these treatments (ext. fig. 4b and ext. fig. 5c), making it unlikely to be PI3P-dependent. In particular, the claim "VPS34 inhibitor(VPS34i) treatment strongly reduced the recruitment of RUFY3 and clustering of LAMP1+ EL in EBSS starved cells (Extended Data Fig. 4a & 4b)." (page7) should be revised as the localization of iRUFY3 at the perinuclear area actually seems unaffected by the treatment.

The authors wrote, "When the PIKfyve inhibitor (YM201636) was used, a strong association of iRUFY3 with large tubules emanating from ELs and enriched in PtdIns(3)P34 was observed at steady state and more prominently in EBSS starved cells (Extended Data Fig. 4c & 5a-b)." (page 7) However, there is no PI3P marker on these figures. The enrichment of the tubule for PI3P and colocalization with iRUFY3 must be shown with a 2xFYVE construct or antibody under the different treatments.

Such a construct is used in Ext Fig 5d and could bring critical insights. However better quality of the dataset is required and the design of the figure must be revised. The perinuclear localization of iRUFY3 is not visible here and the 2xFYVE signal is now more peripheric. What is the caption for the top panel? What do cell and cortical mean and how do the authors interpret it? The effect of Torin on 2xFYVE GFP staining should be shown (microscopy images).

The authors wrote, "iRUFY3+ tubules were found in close vicinity to RAB11a+ RE, suggesting that iRUFY3 could favor interactions between different endosome subsets upon PtdIns(3)P accumulation." As it stands, this observation and suggestion appear farfetched and not supported by the data.

3_Cell migration. The authors provide interesting insight on the impact of iRUFY3 on cell migration and propose that the role of iRUFY3 in endosome positioning would be what impact cell migration it rufy3-/- cells (the authors wrote, "iRUFY3-dependent ELs positioning seems therefore critical for cell migration, independently of major cytoskeleton reorganization"). However, this is not shown by the data. For instance, the authors should perform Alr8b KD in WT and Rufy3-/- cells and check migration/ wound healing. In Ext Fig 7a-b, a rescue condition should be included.

4_Salmonella. The authors study the impact of iRufy3 during Salmonella infection of immune cells. Some elements of the literature are not correctly introduced. The authors wrote, "The interaction of ARL8-GTP with HOPS promotes fusion of lysosomes with late endosomes and autophagosomes, and Salmonella-containing vacuoles, in some cases in cooperation with PLEKHM1 and PLEKHM219,20,21." To my knowledge, there is no evidence of the involvement of the interaction between Arl8b and HOPS in the SCV-lysosome fusion in the mentioned references (or elsewhere as far as I know – please correct me if I am wrong). The reference on Salmonella and Arl8b to use here is Ref#37 which shows that Arl8b is associated with Sif formation and late (20hpi) perifugal displacement of Salmonella for cell-to-cell transfer. In the text, "SCV formation" should be replaced by "SCV maturation" which refers more to the establishment of a viable bacterial niche at later timepoints of the infection.

The authors wrote, "showing that iRUFY3 is not involved with bacterial phagocytosis (Fig. 4c),

although it rapidly localizes to the SCV containing GFP-Salmonella, LAMP1 and the bacterial effector PipB238 (Fig. 4d and 4e)." The images are taken at 16 hpi, so the authors cannot claim that iRUFY3 localizes rapidly to the SCV.

Fig 4d and e: for a better comparison of the localization of RUFY3 at the SCV, the authors should use an earlier timepoint and/or the MOI must be adapted to see only a few bacteria per cell. Now the images show cells filled up with bacteria, making it difficult to interpret protein recruitment.

The authors wrote, "Interestingly, iRUFY3 recruitment to the SCV prevents perinuclear positioning of iRUFY3+/LAMP1+ ELs in EBSS-starved infected macrophages (Fig. 4e)." There is no proof that it is iRUFY3 recruitment at the SCV that disrupt the perinuclear positioning of LAMP1+ ELs. Of note, LAMP1 is also recruited at SCV. I suggest removing this claim.

Minor comments:

- the naming of the different variants of Ruffy3 differs in the literature. (ruffy3xl, etc). Here I believe that iRuffy3 and nRuffy3 are for immune cells and neuronal expression, respectively. It may be worth mentioning it somewhere in the text. Also, I recommend always precisising the isoform (i or n) everywhere possible (apart from when referring to the gene, obviously).
- The manuscript is well written but the navigation between the different (extended) figure panels makes the result section hard to follow. The difference between the main and extended figures is sometimes unclear in terms of importance. Some of the panels presented appears redundant/unnecessary and may overall dilute the message of the most relevant experiments (for instance Ext Fig 4a vs Fig 2d). Idem for the 3D modeling in Ext. Fig 2 (actually the clustering in ext fig2b-LPS is less clear than in Fig2c, quantification of the clustering phenotype would be more useful). Besides, microscopy images are not always easy to read/interpret. It would help to have the cell shape outlines in dash lines.
- Fig1i: the result of a 2.5 increase at the RNA level but a non-significant 1.2 increase at the protein level is surprising. Could the authors provide some explanation? Besides, why is there no error bar? Please include the dots of individual experiments in Fig. 1i and in Ext. Fig 1f.
- Ext. fig 2c, LPS treatment: The cells seem zoomed in or of a very different size than cells displayed in steady state and EBSS treatment. The authors should rather provide comparable cells.
- Ext. fig 4c, rab11 panel should also show iRUFY3+ tubules to check if rab11 localize of these tubules.
- The authors wrote, "The ruffy3 gene was inactivated using Crispr/Cas9 technology in RAW macrophages (ruffy3 -/-), prior reconstitution to near physiological levels with a Myc-tagged version of each isoform (RAW-iRUFY3 and RAW-nRUFY3) and comparative analysis of their respective functions (Fig. 1d)." The words "near physiological" should be removed or better justify, in particular as nRUFY3 is not physiologically expressed in these cells.
- The title of y-axes could often be more informative (for instance Fig 1f, 4b, 4c).
- Some terms appearing on the figures should be explained in the caption. For instance, MFI/gMFI in y-axes, " ns" (non-specific), W and K in Fig 5a, etc.
- Fig 4d right panel, some band appears in the bottom left corner suggesting (harmless) image manipulation. This should be corrected.
- Ext Fig 6d. Does the HeLa band correspond to protein precipitate (explaining the absence of actin bands)? This should be in the caption
- The authors wrote, "Upon Flt3L-DC, MuTu DC or alveolar macrophages activation with different MAMPs, iRuffy3 was found to be the only RUFY family member to be up-regulated transcriptionally after 8 hours of stimulation (Fig. 1f and Extended Data Fig. 1b & 1f)." Was the RNA seq targeting

iRUFY3 specifically? If yes, the caption and figure words of Fig 1F should be changed. Idem for Ext Fig 1f. Otherwise, the main text should be changed (iRUFY3 -> rufy3).

- The result and figure titles "iRufy3 deletion alters antigen processing and presentation in APCs" appears contradictory with the conclusion that "some key EL functions like IFN-g signaling, antigen processing and MHC II-restricted presentation are enhanced in activated rufy3 -/- RAW cells." This should be corrected

Additional suggestions:

- Discussion: The authors may rise the point of the mechanism controlling tissue-dependent alternative splicing of rufy3.

- Structure: The authors may consider changing the order of the figures and results to improve the flow.

Reviewer #3 (Remarks to the Author):

In this work, the authors report that the FYVE domain-bearing isoform designated iRUFY3, which is principally expressed by key innate immune cells and is up-regulated by PAMPS, is necessary for ARL8b+/LAMP1+ endo-lysosome positioning to the pericentriolar cloud. This is an interesting set of observations based on convincing (for the most part, see comments below) confocal microscopy-based data.

The authors go on to carry out in vitro experiments with the goal of demonstrating that iRUFY3 plays a role in multiple macrophage/DC functions including responsiveness to IFN γ , trafficking, MHC II expression, antigen presentation, and intracellular pathogen survival. In general these experiments yielded interesting findings. However the attempts to demonstrate in vivo significance for selective deficiency of iRUFY3 in CD11c+ macrophages and DCs were much less compelling calling into question the organism-level significance of iRUFY3 in macrophage/DC/recruited monocyte function. The conclusions that iRUFY3 has a role in promoting intracellular replication of Salmonella in phagocytic cells and that inactivation of rufy3 in phagocytes leads to aggravated pathologies in mouse upon LPS injection or bacterial pneumonia are not justified by the data provided. Thus the main finding that iRUFY3 is a novel modulator of inflammation that controls endo-lysosomes dynamics, and ultimately regulating the function of activated phagocytes is not robustly supported.

Specific Comments

1. General comment – The manner in which the authors have organized the figures/panels as it relates to the text results in the reader spending an inordinate and frustrating amount of time searching for the appropriate panels thus making the paper unnecessarily difficult to evaluate.
2. General comment – In general, the figure captions require additional detail in order for the reader to understand and evaluate the data.
3. General comment – The methods, especially those provided for the bacterial challenge experiments, lack important details needed for the interpretation of the relevance of the results.
4. Figure 1c – Several pieces of data in Figure 1 and ED Figure 1 indicate the rufy3 gene expression and iRUFY3 protein production is induced by LPS treatment. The data in panel 1c seems to indicate the opposite, especially for the spleen. LPS exposure results in down regulation of iRUFy3 protein. This issue needs to be clarified and/or discussed.
5. Page 6 and Figure 2 – The authors use high resolution immunofluorescence confocal microscopy to localize myc-tagged versions of iRUFY3 and uRUFY3 in the pericentriolar environment of Rufy3-

/- RAW cells. To conclude that 'iRUFY3 mostly associates with LAMP1+ ELs in the pericentriolar area and its FYVE domain is determinant for this localization.' While the microscopy imaging convincingly shows that iRUFY3 appears to preferentially localize to the pericentriolar microenvironment of the cell, the data that iRUFY3 associates with LAMP1+ ELs is significantly less convincing. To add rigor to the claim of LAMP1+ EL association, the authors should consider subcellular fractionation, proximity ligation, or pull-down approaches to generate molecular-level data that can validate the conclusions based on the confocal microscopy findings.

6. Pages 7/8 – While the data used in Figures 4 and 5 to support the conclusion that iRUFY3's noncanonical FYVE domain binds to PtdIns(3)P shows a reasonably strong correlation based on 2D and 3D colocalization of iRUFY3 with putative PtdIns(3)P-containing organelles/structures, this correlation could be due to other molecular interactions. If the authors want to rigorously explore this issue, they should consider designing experiments that can effectively demonstrate direct binding of iRUFY3 to PtdIns(3)P.

7. Page 8 – Provide a rationale for why there is a 40% reduction in Salmonella replication in rufy3-/- RAW cells. The data does not support the conclusion that iRUFY3 plays a role in promoting intracellular replication of Salmonella in phagocytic cells.

8. Page 11 and Figure 9 – This attempt to place biological significance for iRUFY3 deficiency in CD11c+ cells using an E. coli/lung challenge model is flawed at several levels. The protocol is not adequately described in the Methods section and the figure captions do not describe the abbreviations adequately so it is difficult to determine which cell types are being described. At the organismal level, there was no significant difference in the morbidity or mortality between the CD11ccre rufy3fl/fl mice and control animals. The authors might consider exploring other challenge models to discover a phenotype that could provide insights into the mechanism of action of iRUFY3 in CD11c+ cells.

9. Does iRUFY3 come up in any of the human GWAS data sets as being associated with any disease condition?

Point to point answers to the reviewers comments on Char et al.

We thank all the reviewers for their positive comments on our manuscript and their constructive criticisms. We have now address most of these comments and performed new experiments when needed. We hope that these modifications will answer most of their original concerns.

Reviewer #1 (Remarks to the Author):

In this manuscript, Char et al. highlight the function of the FYVE domain-containing longer isoform (iRUFY3) of mouse rufy3, which is preferentially expressed in immune cells, in regulating the positioning of endolysosomes (ELs) in the perinuclear cloud/region of the cells as well as in regulating cell migration, intracellular Salmonella replication, and antigen presentation. Here, the authors first provide evidence that iRUFY3 is specifically expressed in bone marrow, lymphoid organs, and immune cells (including mouse macrophage cell line-RAW264.7), while the canonical shorter rufy3 isoform (nRUFY3) is mostly expressed in the brain. They show that the expression of iRUFY3 is upregulated upon treatment with different MAMPs, IFN α , as well as upon nutrient starvation. Using reconstitution of CRISPR edited RAW cells lacking RUFY3 expression (RAW RUFY3 KO cells) with tagged-forms of iRUFY3 and nRUFY3, iRUFY3 was shown to localize to LAMP1-positive ELs, while nRUFY3 displayed mainly cytosolic distribution. The authors also showed that iRUFY3 localization to LAMP1/lysosomal GTPase Arl8b-positive ELs was increased upon LPS stimulation (and also under the starvation condition) and that iRUFY3 promoted the perinuclear clustering of these cellular compartments. The authors concluded that RUFY3 is required for maintaining the perinuclear pool of ELs through its interaction with Arl8b and promotes retrograde transport of ELs. The authors then focus on the role of RUFY3 in regulating the replication of intracellular pathogens, using Salmonella as a model organism. As previously reported, perinuclear positioning of Salmonella containing vacuoles (SCVs) is important for the development of replicative niches. In the absence of iRUFY3, Salmonella replication was less in RAW macrophages. To elucidate the function of RUFY3 in vivo, the authors generated a transgenic mouse model lacking expression of RUFY3 in CD11c-expressing cells, including most DC subsets and alveolar macrophages. Interestingly, the rufy3-deficient GM-CSF-DCs (as well as RAW macrophages) were less migratory compared to wild-type cells when challenged with LPS, and this was not due to alteration in the actin cytoskeleton. The authors further characterized the CD11c+ cell compartment in a rufy3-deficient mouse model and found it to exhibit a more pro-inflammatory phenotype. This was also in line with higher levels of MHC class II molecules on the surface and better presentation of exogenous antigens.

The manuscript is filled with a ton of data on iRUFY3 function, and with a few exceptions described below, the data is largely convincing and well done. The antigen presentation experiment showing that iRUFY3 depletion interferes with MHC class II surface levels and exogenous antigen presentation in Fig. 5 is interesting and needs further mechanistic insights for a solid conclusion regarding its physiological significance.

While the data presented is interesting, it is mostly descriptive of different phenotypes observed upon iRUFY3 depletion in macrophages. For instance, the authors have made very interesting observations that RUFY3 regulates macrophage cell migration and exogenous antigen presentation. However, the authors have not explored the mechanistic basis for these phenotypes. The manuscript in its present form is incomplete, and the following points warrant attention prior to publication:

We thank reviewer 1 for its positive and constructive comments and have addressed most the concerns raised by this reviewer either by correcting the text or the figures, either experimentally. We have, in particular: a) Evaluated the contribution of ARL8b to iRUFY3 recruitment on ELs using ARL8b silencing, b) Evaluated the rescue potential of the different RUFY3 isoforms on the migration deficit observed in *rufy3*^{-/-} RAW and c) Examined MHC II-associated Invariant chain processing in *rufy3*^{-/-} RAW.

Major points:

1. In the text on page 4, the authors describe the main difference between the nRUFY3 and iRUFY3 variants of the rufy3 gene as the presence of an extended c-terminal region of 150 amino acids containing the FYVE domain. This is a bit confusing. As the difference between the variants is actually 200 amino acids and not 150 amino acids, that is what is shown in Fig. 1a. Are the authors sure that the nRUFY3 referred to here is NP_001346137.1 and not NP_001276704.1, which is 519 a.a. long? This information needs to be clarified and accordingly changed in the text.

The figure was mistakenly labelled and is now corrected with the right number of amino acids, the discrepancy came from a confusion between the different mouse and human rufy3 transcripts.

2. There is variation in the size (kDa) of the iRUFY3 band in several western blots, for example, Figs. 1c/d to 1e/h/i and extended Fig. 1d and 1e. Can the authors clarify the reason for this variation in mobility of the iRUFY3 protein band? Is it possible that it is due to differences in different PTMs in different cell types? This needs to be clarified.

We did not observe evidence for PTM of RUFY3 in the different cell types studied. As expected, a light difference of migration is observed with the “myc” tagged forms of iRUFY3 in Fig 1c/1d compared to the WT RUFY3 expressed in control DCs or tissues. This difference is amplified by the smeary detection of overexpressed positive tagged-iRUFY3 controls, but we did not observe any mobility shift in the different primary cells submitted to different treatments like LPS activation (e.g. control DCs in Ext data Fig 4d).

3. In Fig. 1c and the extended Fig. 6d, why is the b-actin signal missing in lanes in which HeLa cell lysate expressing nRUFY3 and iRUFY3 is loaded? This is a bit surprising, and no explanation is provided in the text or figure legends.

Given the strong overexpression of tagged-i/nRUFY3 positive controls, we had to load minimal amount of control lysates to have an acceptable signal compatible with the detection of RUFY3 in different tissues and primary cells. Unfortunately this minimal loading of transfected HeLa control samples was not sufficient to detect actin with an exposure time compatible with the linear detection of actin present in the other tissue samples. We have indicated this fact in the legend and provide for this reviewer eyes the same blot with longer time of exposure revealing the difference of actin concentration in the different samples. Given that we only used the HeLa samples as MW references for iRUFY3 and nRUFY3, the absence of visible loading control is not influencing the interpretation of the results.

4. What is the rationale for doing the experiment shown in Fig. 1e? Why was LPS treatment given to check the dimerization of iRUFY3? It's already reported in a recent study (Kumar G, Chawla P, Dhiman N, Chadha S, Sharma S, Sethi K, Sharma M, Tuli A. RUFY3 links Arl8b and JIP4-Dynein complex to regulate lysosome size and positioning. *Nat Commun.* 2022 Mar 21;13(1):1540. doi: 10.1038/s41467-022-29077-y. PMID: 35314681; PMCID: PMC8938454) that the CC domain of RUFY3 is responsible for dimerization. However, the schematic depiction of the dimeric form of iRUFY3 in Fig. 1e reflects the involvement of the RUN domain in dimerization, although no experimental evidence is provided to support this claim. However, the authors hypothesize in the result section about the involvement of the CC domain in the dimerization of iRUFY3. The authors should either revise the schematic or provide experimental evidence for the involvement of the RUN domain in dimerization.

After careful examination, we did not find in the cited reference and associated paper, experimental data demonstrating that the CC domains drives RUFY3 dimerization. Although, this is anticipated from the presence of CC domains in the predicted RUFY3 structure, we believe to bring the first biochemical evidence of RUFY3 dimerization. Given that RUFY3 levels are sensitive to TLR activation, it seemed appropriate to test whether its dimerization state could be influenced by LPS activation. However, we agree in the misleading nature of the schematic depiction in Figure 1e, which was corrected and remains neutral concerning the involvement of the specific domains.

5. In general, there is a lot of discrepancy between what is mentioned in the text and what is described in the figure legends. Just an example. In the text on page 5, the authors report an increase in the levels of iRUFY3 in BMDMs upon LPS treatment. However, in the legend of Fig. 1i, the cell type mentioned is RAW macrophages. Firstly, the authors should correct such a mistake. Secondly, if the experiment was performed in BMDMs and not RAW macrophages, then what happens to iRUFY3 levels in RAW macrophages upon treatment with LPS and with other MAMPs treatments?

We apologize for the different errors, unfortunately the wealth of data that we have included in the paper has resulted in some imprecisions or inversions during the different rounds of redaction. We have now corrected Fig 1i and added data corresponding to RAW macrophages.

6. The authors state and illustrate in Figs. 1h and 1i that LPS stimulation induces or augments iRUFY3 expression (or with other MAMP treatments). However, Figs. 1c and 1e show a reduction in the band intensity of iRUFY3 in response to LPS stimulation. These experiments need quantification of blots to better illustrate the claims.

Figures 1c and 1e, cannot be compared to 1h and 1i. Figure 1c shows tissue extracts isolated from mouse submitted or not to LPS challenge. This samples are formed of a mixture of cells with many not expressing RUFY3 and containing obviously different proportions of immune cells, especially after LPS challenge. Figure 1c can only be used to monitor qualitatively iRUFY3 or nRUFY3 expression in different organs, but it is in no way suitable for absolute quantification relative to actin levels given immune cell type heterogeneity in the samples across conditions.

Fig 1e is showing the dimerization of RUFY3 in a qualitative manner, but was not intended to follow quantitatively RUFY3 expression given the relative variation in the response of the samples to non-denaturing conditions. We feel that Figure 1g and 1h demonstrate clearly the up-regulation of RUFY3 by MAMPs in primary cells. However, we observed in parallel a

moderate decrease of RUFY3 protein levels in RAWs macrophages (Fig 1i), which is now commented in the main text of the article.

7. In extended Fig. 1c, there is a ~50% drop in the iRUFY3 mRNA fold change (compare iRUFY3+nRUFY3 to iRUFY3 alone). However, as per the data provided in Fig. 1, there should be no reduction since iRUFY3 is the only form found in DCs. The authors should provide an explanation for this observation.

Given that we used different pairs of primers (with various affinity) to amplify specifically the 2 mRNA targets, the amplification results were not always consistent and their representation was not self-explanatory. We provide in a new Extended Figure 1a, novel results organized in a simplified and readable way, showing that iRUFY3 and nRUFY3 mRNAs expression is mutually exclusive.

8. In Fig. 2 and extended Fig. 2, the author shows that iRUFY3 is recruited to pericentriolar LAMP1+ endosomes in response to LPS activation or upon nutrient deprivation. To substantiate this claim, authors should conduct biochemical experiments (such as membrane cytosol fraction or EL separation) following LPS treatment and starvation. In addition, the rationale for six hours of starvation with EBSS is unclear. A short period of starvation is not enough for the localization of iRUFY3 to LAMP1-positive compartments.

We now provide in new Figure 3e, the results of cell fractionation experiments showing that RUFY3 is mostly cytosolic and is recruited to organelles upon LPS stimulation.

6h of EBSS starvation is the time which yields the most contrasting results in terms of pericentriolar localization of the LAMP1+ compartment and we have therefore kept this time for experimental homogeneity across the paper.

9. For the Pearson coefficient quantification graphs shown in Fig. 2b and extended Figs. 4b and 4c, the authors must describe the whole approach used for analysis in the material and methods section. Furthermore, no information is provided on the number of cells used for quantification, which should be included in all the figure legends. Some samples have a small number of data points, so the authors should increase the number of data points to be consistent.

We have now added the number of cells used in the different quantification, increased these numbers whenever needed and described the approach in the material and methods section.

10. To determine the increase in iRUFY3 recruitment on EL or other organelles, it is important to quantify (Mander's coefficient) what fractions of EL or other organelles colocalize with iRUFY3 upon LPS treatment and starvation in comparison to untreated cells.

We have now applied Manders' Overlap Coefficient (MOC) as a measure of colocalization and an associated scatterplot is displayed in Figure 2b and other figures across the manuscript as well as commented in the text.

11. Both starvation and LPS treatment induce EL clustering in the juxta-nuclear region, and a smudge is visible in the majority of confocal micrographs. Therefore, quantifying protein colocalization under such conditions is challenging. To get a better idea of how much more iRUFY3 is being recruited to ELs, the authors should do these experiments in the presence of microtubule polymerizing inhibitors.

We feel that the 3D reconstitution provided in Fig. 3 (former Extended Data Fig 2) and the biochemical results added in this new version both confirm and support our conclusions. We have nevertheless used Nocodazole to further confirm the association of RUFY3 to LAMP1+ compartments and their dependence on MT for their pericentriolar location (new Extended Data Fig. 2a).

12. In Fig. 3, what do the dots in the various graphs represent? There is no information provided in the figure legend or in the M&M section. To strengthen the PLA data, the authors should biochemically (co-ip) test the interaction between Arl8b and nRUFY3 and iRUFY3. Also, what happens to iRUFY3 localization in cells lacking Arl8b at steady state, during LPS stimulation, and upon starvation? Will it depict cytosolic distribution like nRUFY3 in cells lacking Arl8b? Only the FYVE domain of iRUFY3 is solely sufficient for perinuclear clustering of ELs?

The graphs and legends in Figure 3 have now been completed and required information provided. The symmetrical co-IP of overexpressed ARL8b and RUFY3 was shown in the work of Keren-Kaplan et al., but we failed to pull down ARL8b by performing a iRUFY3 IP on RAW cells expressing normal levels of the two proteins. However, results of new ARL8b knock-down experiments (new Figure 5 and extended Fig. 3) confirm that RUFY3 is not only dependent on ARL8b for its recruitment to EL, but also for its expression.

13. For the experiments shown in extended Figs. 4 and 5, the rationale for including the EBSS treatment along with YM201636 or VPS34i is not clear. From the graph shown in extended Fig. 4c, it's not clear why iRUFY3 colocalization is decreased with LAMP1 under the condition of YM201636 treatment? The authors should clarify this point.

The rationale is that upon nutrient starvation, VPS34 is activated to produce PI3P, EBSS co-treatment with YM201636 or VPS34i should therefore maximize their inhibitory effect and potential impact on RUFY3 distribution. Upon YM201636 treatment RUFY3 accumulates on long tubules connected to LE but depleted of LAMP1, as observed in the 3D rendering in new extended data Fig. 3. In response to reviewer 2/3 comments. we have now also shown that differently from its overexpression in HeLa cells, iRUFY3 is not targeted to PtdIns(3)P enriched domain decorated with the GFP-FYVE probe in RAW cells (New Fig. 6 and Extended data Fig. 3e), suggesting that the effect of PtdIns(3)P accumulation on RUFY3 is indirect. (see also answer to reviewer 3).

14. The experiment shown in Fig. 4a is difficult to comprehend, and the author's assertion about the increase in the quantity and volume of ELs based on this one experiment necessitates further investigation. There is no correlation between the increase in EL number and volume shown in this experiment and the confocal images and staining of LAMP1 and LAMP2 in Figs. 2 and 3. Furthermore, no information on how this experiment performed is provided in the M&M section.

Given that size of the data set presented in the paper was found plethoric, we decided to remove the results presented in old Fig. 4a, since they are not critical to support our observations and the main conclusion of the paper, although they are similar to the findings of Kumar et al. in HeLa cells.

15. In Fig. 4d, the authors show the recruitment of iRUFY3 around the SCV and suggest that iRUFY3 participates in the maturation of the SCV. However, no experiment has been conducted to corroborate this observation. The authors should check the positioning of SCVs in RUFY3 KO cells. Also, it should be determined if the recruitment of iRufy3 to SCVs is dependent on or independent of Arl8b. Again, the rationale for combining starvation with a Salmonella infection is not clear.

We have corrected the text to match the presented data. Given the results obtained after ARL8b silencing presented in the new Figure 5, we have no reason to believe that it would be different for the iRUFY3 recruitment to the SCV.

16. The role of RUFY3 in regulating cell migration is interesting, but it's not thoroughly investigated. The authors can decide to remove this data set or else should investigate in more detail by looking at the staining of focal adhesions, integrins surface levels, etc. in RUFY3 KO cells. Though the authors claim that they do not see any major change in the actin cytoskeleton in RUFY3 KO cells, the number of filopodia seems to be higher in RUFY3 KO cells based on the confocal micrographs shown in extended Fig. 7c.

Given the breadth of the paper, we feel that further exploring this functional aspect of iRUFY3 biology will be truly overwhelming and we would rather perform these additional experiments in a follow-up work. We however provided a complementation experiment with iRUFY3 and nRUFY3 to establish whether their role in regulating macrophage migration could be redundant or not, therefore giving a hint on their potential mechanisms of action. We do not share the impression of this reviewer concerning an increased number of filopodia in RUFY3 $-/-$ cells, but we removed this non critical figure from the manuscript to avoid further confusion. We think however that reporting the migration deficiency of RUFY3 $-/-$ cells is absolutely necessary to partially explain the immunophenotypes of RUFY3 $-/-$ mouse after LPS injection in vivo.

17. The effect of rufy3 depletion on antigen processing and presentation by APCs is very interesting but not mechanistically explored by the authors. The authors should test if the lack of iRUFY3 effects the recycling rate of peptide-bound MHC-II since, in the absence of iRUFY3, ELs are positioned closer to the plasma membrane. Such experiments will provide better insights for iRUFY3 role in immune cells and will also be useful to comprehend the pro-inflammatory phenotype of the CD11c+ compartment observed in the mouse model of RUFY3 KO as shown in Fig. 6 and extended Figs. 8 and 9.

We have further characterized the MHC II transport in RUFY3 $-/-$ RAWs by investigating Invariant chain (Ii) after IFN γ treatment. RUFY3 deletion clearly increases Ii processing and flux suggesting that it contributes greatly to the enhancement of antigen presentation upon IFN exposure. However exploring further the details MHC II trafficking in absence of iRUFY3, will be again truly overwhelming and should rather be the basis of a novel follow-up project performed both in macrophages and dendritic cells.

18. The figure legend for extended Fig. 10 needs to be described in detail.

The legend was corrected.

19. There have been two recent publications on RUFY3 as an ARL8b effector and how the RUFY3-ARL8b complex regulates perinuclear lysosome positioning (PMID: 35314681 and 35314674). Though the authors have cited these publications in the manuscript, at some places one or the other work is not cited. Since the results shown in both of these studies are complementary and these studies came out at the same time, it is important to give equal credit to both research groups. Therefore, the two publications should be cited together wherever they are referred to in the manuscript text.

It was not our intent to separate these two important studies. Although very complementary, they also present some individual observations, sometimes justifying different citations (e.g. RUFY4 interaction with ARL8b is only demonstrated in Keren-Kaplan et al.). We have however rewrote the text to have the two papers cited together and mention adequately their results.

Minor points:

Note: There are mistakes as well as missing information in the manuscript text, figure labels, and figure legends.

1. The labeling of "RAW 264.7" cells should be kept consistent both in manuscript text, figure labels, and legends.

This was corrected.

2. RUFY3 CRISPR-edited knockout cells should be mentioned as "RAW RUFY3^{-/-}" instead of "RAW^{-/-}." This information should be corrected in the manuscript text, figure labels, and figure legends.

This was corrected.

3. The evidence of RUFY3 KO is provided as a series of immunoblots that lack detectable protein. Given that the KO cells are single-cell clones, the authors should also provide a summary of their sequencing data to show that the *rufy3* gene was disrupted.

We have now provided in Extended Figure 1i, the PCR demonstration of effective RUFY3 gene disruption in the different selected clones.

4. "Fig. 1l" does not exist as mentioned in the text; it should be changed to "Fig. 1i."

This was corrected.

5. In manuscript text, "zinc finger motifs" is mistyped as "zing finger motives." Also, "nutrient starvation" is mistyped as "nutriment starvation".

This was corrected.

6. In Fig. 3b (the right-side graph), what the blue and white bars represent is not labeled.

This was corrected.

7. In the extended Fig. 5d graph, the labeling should be aligned properly.

This figure was removed.

8. The figure legend for Fig. 5e is lacking information on what the dot represents.

This was corrected.

9. The figure legends lack information regarding certain elements, such as Fig. 1c (MLN and ILN) and Fig. 1e (W and K).

This was corrected.

Reviewer #2 (Remarks to the Author):

In this study, Char and colleagues explore the expression and functions of iRUFY3, a rufy3 splicing variant, focusing on immunity. rufy3 has 2 splicing variants: nRUFY3 (without FYVE domain) and iRUFY3 (with a FYVE domain). While nRUFY3 interacts with the actin network and has been mostly studied in the context of nervous system development, the iRUFY3 isoform has recently dragged attention in a back-to-back publication in Nat. Comm. (Kumar et al., 2022; Keren-Kaplan et al., 2022) showing that iRUFY3 interacts with Arl8b to position endosomes.

Here, Char and colleagues provide more insights into the tissue-specific expression of iRUFY3. Following their observation of a preferential expression of iRUFY3 in (activated) immune cells, the authors investigate different functions of iRUFY3 in immunity. They observe that the endosome clustering in LPS-activated macrophages is iRUFY3 dependent, while the migration of macrophages and DC as well as antigen presentation in activated cells is reduced by iRUFY3. Besides, iRUFY3 contributes to Salmonella intracellular replication in macrophages, potentially by contributing to the formation of a mature/viable bacterial vacuolar niche. Finally, the deletion of rufy3 appears pro-inflammatory, suggesting an overall versatile function of iRUFY3 in modulating immune reactions.

This study explores many aspects of iRUFY3 and provides several novel and relevant insights, in particular on rufy3 tissue-dependent alternative splicing and iRufy3 roles in immune cells. However, the different directions of the paper are not always connected – at least at the experimental level. As a result, the proposed model and the title assume many links that are not (yet) supported by the data. Besides, the part of the manuscript on PI3P is not convincing and requires more in-depth investigation.

Major comments:

1_ General. As currently written, the introduction and some elements of the results undermine the previously published work showing that iRUFY3 interacts with Arl8b to regulate lysosome size and positioning (Kumar et al., 2022; Keren-Kaplan et al., 2022). Although the reference is indeed quoted, the overlap with some of the claims of the paper is not transparently exposed, which could wrongly suggest that the interaction of iRUFY with Arl8b and its role in endosome positioning is a novel finding from Char et al. I would recommend the authors to be more upfront on what has been published and what is the adding of their work. This will help the readers to focus on the key findings of this manuscript.

For instance:

- in the sentence “several effectors have been identified for mammalian ARL8, [...] and recently the RUN and FYVE domain-containing protein 3 (RUFY3)” (page 3), RUFY3 should be replaced by iRUFY3 to be clear that this is the same isoform as in this study.

- in the sentence “iRUFY3 drives the pericentriolar clustering of ARL8b/LAMP1+ ELs and is closely linked to PtdIns(3)P production.” (page 4), as these elements were previously published, references should be added.

It was not our intent to diminish the importance of these two studies, we have now modified the text and cited when needed the findings of our colleagues. We also have changed the title to follow this reviewer’s concern and further support some of our findings.

2_ iRufy3 endosomal localization and PI3P+ compartments. The authors display many microscopy panels in the main and extended figures showing the endosomal localization of iRUFY3 in immune cells upon different treatments (LPS, EBSS) (see also minor points). The authors wrote: “Although sorting and recycling endosomes (SE, RE) markers EEA1, RAB11A and Syntaxin 6 were found in close vicinity with iRUFY3, the late endosomal marker LAMP1 displayed the best co-localization with iRUFY3 (Fig. 2b).” But actually, EEA1 is not visibly in close vicinity with iRUFY3 on any of the 3 tested conditions. This is an important point as EEA1 is the textbook definition of a PI3P-bound protein, which is less expected for LAMP1.

We fully agree on the importance of this point and have corrected the text about EEA1 positioning accordingly (p6).

Then the authors explore the FYVE-domain-dependent localization of iFURY3 on PI3P+ compartments. For this, they use PIKFyve and VPS34 inhibitors to change the internal pool of PI3P. The impact of PIKFyve and VPS34 in PI3P should be better explained in the text.

ie, VPS34 phosphorylate PI into PI3P and PIKFyve phosphorylate PI3P into PI3,5P2, making the inhibition of VPS34 and PIKFyve decreases or increases intracellular PI3P respectively. The result should be interpreted/discussed along that line. In particular, the localization of iFURY3 does not seem to be affected by these treatments (ext. fig. 4b and ext. fig. 5c), making it unlikely to be PI3P-dependent. In particular, the claim “VPS34 inhibitor(VPS34i) treatment strongly reduced the recruitment of RUFY3 and clustering of LAMP1+ EL in EBSS starved cells (Extended Data Fig. 4a & 4b).” (page7) should be revised as the localization of iRUFY3 at the perinuclear area actually seems unaffected by the treatment.

We have now clarify the text and described better the effect of the inhibitors. We understand the remarks of this reviewer and amended the text accordingly to be more precise. We never claimed that interacting with PtdIns(3)P is the sole manner of recruiting iRUFY3 to LAMP1+ LE, as we, and others, have now shown the implication of ARL8b in this process. However, we observed that upon VPS34i treatment, iRUFY3 does not co-localize with LAMP1+LE, which do not concentrate anymore in the perinuclear area. Therefore our claim is accurate and PtdIns(3)P is required for the RUFY3-dependent positioning of LAMP1+ LE in the perinuclear zone. As stated by reviewer 2, under these conditions iRUFY3 is still found accumulating in the pericentriolar area independently of LAMP1+ organelles. We have currently no explanation for that phenomenon, however given its known interaction with ARL8b and dynein, iRUFY3 could be transported independently of VPS34 activity and PtdIns(3)P formation, in molecular complexes or through affinity with other PtdIns bearing organelles. Importantly, we have now shown that iRUFY3 does not directly interact with PtdIns(3)P enriched domains decorated with 2xFYVE GFP, in RAWs, suggesting that the activity of the different inhibitors on RUFY3 recruitment to ELs are likely to be indirect, rather than through PtdIns(3)P binding. It is however clear that iRUFY3 has affinity for membrane organelles independently of its binding to ARL8b, as suggested by the results of the different KD experiments presented in a new fig. 5. We have now changed accordingly the text of the paper commenting these results.

The authors wrote, “When the PIKfyve inhibitor (YM201636) was used, a strong association of iRUFY3 with large tubules emanating from ELs and enriched in PtdIns(3)P34 was observed at steady state and more prominently in EBSS starved cells (Extended Data Fig. 4c & 5a-b).” (page 7) However, there is no PI3P marker on these figures. The enrichment of the tubule for PI3P and colocalization with iRUFY3 must be shown with a 2xFYVE construct or antibody under the different treatments.

Such a construct is used in Ext Fig 5d and could bring critical insights. However better quality of the dataset is required and the design of the figure must be revised. The perinuclear localization of iRUFY3 is not visible here and the 2xFYVE signal is now more peripheric. What is the caption for the top panel? What do cell and cortical mean and how do the authors interpret it? The effect of Torin on 2xFYVE GFP staining should be shown (microscopy images).

We have now corrected this sentence and modify the figures as described above.

The authors wrote, “iRUFY3+ tubules were found in close vicinity to RAB11a+ RE, suggesting that iRUFY3 could favor interactions between different endosome subsets upon PtdIns(3)P accumulation.” As it stands, this observation and suggestion appear farfetched and not supported by the data.

We agree with this reviewer and have now modified the text and removed most of RAB11a data to simplify the reading of the figures and avoid speculations.

3_Cell migration. The authors provide interesting insight on the impact of iRUFY3 on cell migration and propose that the role of iRUFY3 in endosome positioning would be what impact cell migration it rufy3-/- cells (the authors wrote, “iRUFY3-dependent ELs positioning seems therefore critical for cell migration, independently of major cytoskeleton reorganization”). However, this is not shown by the data. For instance, the authors should perform Alr8b KD in WT and Rufy3-/- cells and check migration/ wound healing. In Ext Fig 7a-b, a rescue condition should be included.

ARL8b and RAB7b have been previously associated with migration deficiency in different cell lines. We added complementation data for rufy3-/- RAWs and shown that nRUFY3 is unable to rescue their migration deficit while iRUFY3 does, suggesting that the two isoforms use a different mode of action to regulate migration. We have also shown that binding to the previously described Rap2 and Fascin axis is unlikely to be used by iRUFY3, since Fascin is not expressed by RAW macrophages. In addition, we did not observe obvious change in actin organization at the microscopic level in absence of RUFY3 (data removed as requested by reviewer 1).

4_Salmonella. The authors study the impact of iRufy3 during Salmonella infection of immune cells. Some elements of the literature are not correctly introduced. The authors wrote, “The interaction of ARL8-GTP with HOPS promotes fusion of lysosomes with late endosomes and autophagosomes, and Salmonella-containing vacuoles, in some cases in cooperation with PLEKHM1 and PLEKHM219,20,21.” To my knowledge, there is no evidence of the involvement of the interaction between Arl8b and HOPS in the SCV-lysosome fusion in the mentioned references (or elsewhere as far as I know – please correct me if I am wrong). The reference on Salmonella and Arl8b to use here is Ref#37 which shows that Arl8b is associated with Sif formation and late (20hpi) perifugal displacement of Salmonella for cell-to-cell transfer. In the text, “SCV formation” should be replaced by “SCV maturation” which refers more to the establishment of a viable bacterial niche at later timepoints of the infection.

This was corrected accordingly to these suggestions.

The authors wrote, “showing that iRUFY3 is not involved with bacterial phagocytosis (Fig. 4c), although it rapidly localizes to the SCV containing GFP-Salmonella, LAMP1 and the bacterial effector PipB238 (Fig. 4d and 4e).” The images are taken at 16 hpi, so the authors cannot claim that iRUFY3 localizes rapidly to the SCV.

This was amended.

Fig 4d and e: for a better comparison of the localization of RUFY3 at the SCV, the authors should use an earlier timepoint and/or the MOI must be adapted to see only a few bacteria per cell. Now the images show cells filled up with bacteria, making it difficult to interpret protein recruitment

Given the purely descriptive nature of our findings on the role of iRUFY3 during Salmonella infection, we felt that further entering into the mechanistic of recruitment of iRUFY3 to the SCV was not in the scope of the present manuscript and should be left unexplored for future publications in collaboration with host-pathogen interaction specialists.

The authors wrote, “Interestingly, iRUFY3 recruitment to the SCV prevents perinuclear positioning of iRUFY3+/LAMP1+ ELs in EBSS-starved infected macrophages (Fig. 4e).” There is no proof that it is iRUFY3 recruitment at the SCV that disrupt the perinuclear positioning of LAMP1+ ELs. Of note, LAMP1 is also recruited at SCV. I suggest removing this claim.

This was removed.

Minor comments:

- the naming of the different variants of Ruffy3 differs in the literature. (ruffy3xl, etc). Here I believe that iRuffy3 and nRuffy3 are for immune cells and neuronal expression, respectively. It may be worth mentioning it somewhere in the text. Also, I recommend always precisising the isoform (i or n) everywhere possible (apart from when referring to the gene, obviously).

We fully agree on this remark as we were at the origin of the RUFY3XL naming (Char et al. 2020) and decided to change to iRUFY3 and nRUFY3 given our tissue and cell type expression results. It is now mentioned in the text.

- The manuscript is well written but the navigation between the different (extended) figure panels makes the result section hard to follow. The difference between the main and extended figures is sometimes unclear in terms of importance. Some of the panels presented appears redundant/unnecessary and may overall dilute the message of the most relevant experiments (for instance Ext Fig 4a vs Fig 2d). Idem for the 3D modeling in Ext. Fig 2 (actually the clustering in ext fig2b-LPS is less clear than in Fig2c, quantification of the clustering phenotype would be more useful). Besides, microscopy images are not always easy to read/interpret. It would help to have the cell shape outlines in dash lines.

We have tried to improve the overall readability of the manuscript and the figures by removing some of the less important data and focusing on the most determinants (e.g. for Fig 2 and extended 2). Originally the manuscript was formatted for another Springer press journal and we have now taken full advantage of the Nature Communications format to increase the number of full figures and decrease the amount of navigation with extended data.

- Fig1i: the result of a 2.5 increase at the RNA level but a non-significant 1.2 increase at the protein level is surprising. Could the authors provide some explanation? Besides, why is there no error bar? Please include the dots of individual experiments in Fig. 1i and in Ext. Fig 1f.

It is difficult to give a precise explanation for the non-linear relationship between transcription and translation levels, which is rather common for many transcripts in active phagocytes. In the present case, LPS activation is known the change qualitatively and quantitatively the level of translation during time (Lelouard et al. JCB, 2007), as well as the endosomes dynamics (this work), which could both influence RUFY3 protein homeostasis in the bmDM model, leading to a more moderate increase in RUFY3 levels than expected from its mRNA induction, this differently from the results obtained in the bmDCs model, which are quite clear cut (Fig. 1h).

Figure panels now in Ext Fig 1f and 1 g were modified as requested (dots).

- Ext. fig 2c, LPS treatment: The cells seem zoomed in or of a very different size than cells displayed in steady state and EBSS treatment. The authors should rather provide comparable cells.

This is now Full figure 3 and a new organization is proposed with the cell boundaries indicated.

- Ext. fig 4c, rab11 panel should also show *iRUFY3+* tubules to check if rab11 localize of these tubules.

We have removed most of Rab11 data from most of the different figures since they were complicating the readability by overloading the figures, while not bringing any key information.

- The authors wrote, "The *rufy3* gene was inactivated using Crispr/Cas9 technology in RAW macrophages (*rufy3* -/-), prior reconstitution to near physiological levels with a Myc-tagged version of each isoform (RAW-*iRUFY3* and RAW-*nRUFY3*) and comparative analysis of their respective functions (Fig. 1d)." The words "near physiological" should be removed or better justify, in particular as *nRUFY3* is not physiologically expressed in these cells.

This was corrected.

- The title of y-axes could often be more informative (for instance Fig 1f, 4b, 4c).

This was corrected and some of these graphs removed (old Fig 4b and 4).

- Some terms appearing on the figures should be explained in the caption. For instance, MFI/gMFI in y-axes, "ns" (non-specific), W and K in Fig 5a, etc.

This was corrected.

- Fig 4d right panel, some band appears in the bottom left corner suggesting (harmless) image manipulation. This should be corrected.

This was corrected.

- Ext Fig 6d. Does the HeLa band correspond to protein precipitate (explaining the absence of actin bands)? This should be in the caption.

As stated in the answer to reviewer 1 comments, we have provided an explanation in the figure legend and a more exposed blot for reviewer's eyes.

- The authors wrote, "Upon Flt3L-DC, MuTu DC or alveolar macrophages activation with different MAMPs, *iRufy3* was found to be the only RUFY family member to be up-regulated transcriptionally after 8 hours of stimulation (Fig. 1f and Extended Data Fig. 1b & 1f)." Was the RNA seq targeting *iRUFY3* specifically? If yes, the caption and figure words of Fig 1F should be changed. Idem for Ext Fig 1f. Otherwise, the main text should be changed (*iRUFY3* -> *rufy3*).

This was corrected.

- The result and figure titles "*iRufy3* deletion alters antigen processing and presentation in APCs" appears contradictory with the conclusion that "some key EL functions like IFN-g signaling, antigen processing and MHC II-restricted presentation are enhanced in activated *rufy3* -/- RAW cells." This should be corrected

This was corrected.

Additional suggestions:

- Discussion: The authors may rise the point of the mechanism controlling tissue-dependent alternative splicing of *rufy3*.

We have added a comment on this, but we really do not have much experimental data for this except for the neuron specificity.

- Structure: The authors may consider changing the order of the figures and results to improve the flow.

We believe to have improve the flow of the manuscript by simplifying the writing and removing some data from the figures as well as taking advantage of the Nature Communications format.

Reviewer #3 (Remarks to the Author):

*In this work, the authors report that the FYVE domain-bearing isoform designated *iRUFY3*, which is principally expressed by key innate immune cells and is up-regulated by PAMPs, is necessary for ARL8b+/LAMP1+ endo-lysosome positioning to the pericentriolar cloud. This is an interesting set of observations based on convincing (for the most part, see comments below) confocal microscopy-based data.*

*The authors go on to carry out in vitro experiments with the goal of demonstrating that *iRUFY3* plays a role in multiple macrophage/DC functions including responsiveness to IFN γ , trafficking, MHC II expression, antigen presentation, and intracellular pathogen survival. In general these experiments yielded interesting findings. However the attempts to demonstrate in vivo significance for selective deficiency of *iRUFY3* in CD11c+ macrophages and DCs were much less compelling calling into question the organism-level*

significance of iRUFY3 in macrophage/DC/recruited monocyte function. The conclusions that iRUFY3 has a role in promoting intracellular replication of Salmonella in phagocytic cells and that inactivation of irufy3 in phagocytes leads to aggravated pathologies in mouse upon LPS injection or bacterial pneumonia are not justified by the data provided. Thus the main finding that iRUFY3 is a novel modulator of inflammation that controls endo-lysosomes dynamics, and ultimately regulating the function of activated phagocytes is not robustly supported.

Although the phenotypes observed in the CD11C-RUFY3 flx/flx mouse are not severe, they are clearly reproducible and consistently point towards a pro-inflammatory situation in the whole animal, which also seems to increase with age (not included in the paper). We therefore disagree with this reviewer remark and argue that the important set of data provided in this work supports our claim that:

- a) iRUFY3 that controls endo-lysosomes dynamics,
- b) iRUFY3 is a novel modulator of inflammation and interferes with the function of activated phagocytes.

Specific Comments

1. General comment – The manner in which the authors have organized the figures/panels as it relates to the text results in the reader spending an inordinate and frustrating amount of time searching for the appropriate panels thus making the paper unnecessarily difficult to evaluate.

We have tried to simplify the organization and prioritize key figures in agreement with Nature Communications format as indicated in our answers to the other reviewers.

2. General comment – In general, the figure captions require additional detail in order for the reader to understand and evaluate the data.

We have corrected this in light of other reviewers comments.

3. General comment – The methods, especially those provided for the bacterial challenge experiments, lack important details needed for the interpretation of the relevance of the results.

We have provided additional information about the pneumonia challenge.

4. Figure 1c – Several pieces of data in Figure 1 and ED Figure 1 indicate the irufy3 gene expression and iRUFY3 protein production is induced by LPS treatment. The data in panel 1c seems to indicate the opposite, especially for the spleen. LPS exposure results in down regulation of iRUFy3 protein. This issue needs to be clarified and/or discussed.

See our answer to Reviewer 1's comment number 6 (p 2-3).

5. Page 6 and Figure 2 – The authors use high resolution immunofluorescence confocal microscopy to localize myc-tagged versions of iRUFY3 and uRUFY3 in the pericentriolar environment of Rufy3-/- RAW cells. To conclude that 'iRUFY3 mostly associates with LAMP1+ ELs in the pericentriolar area and its FYVE domain is determinant for this localization.' While the microscopy imaging convincingly shows that iRUFY3 appears to preferentially localize to the pericentriolar microenvironment of the cell, the data that iRUFY3 associates with LAMP1+ ELs is significantly less convincing. To add rigor to the claim of LAMP1+ EL association, the authors should consider subcellular fractionation, proximity ligation, or pull-down approaches to generate molecular-level data that can validate the conclusions based on the confocal microscopy findings.

We have now added new results addressing this concern (New Fig 3e) and see our answer to Reviewer 1's comment number 8 (p 3).

6. Pages 7/8 – While the data used in Figures 4 and 5 to support the conclusion that iRUFY3's noncanonical FYVE domain binds to PtdIns(3)P shows a reasonably strong correlation based on 2D and 3D colocalization of iRUFY3 with putative PtdIns(3)P-containing organelles/structures, this correlation could be due to other molecular interactions. If the authors want to rigorously explore this issue, they should consider designing experiments that can effectively demonstrate direct binding of iRUFY3 to PtdIns(3)P.

Our efforts to use specific PtdIns coated membranes to further confirm RUFY3 binding to PtdIns in general were not successful. Given that this point was raised by different reviewers, we re-evaluated the co-distribution of RUFY3 with the GFP-2-FYVE construct in iRUFY3-RAW cells, and differently from our original data in co-transfected HeLa cells, we could not see any significant staining overlap between the two molecules. We therefore changed our conclusions and amended the text accordingly. We will feel that although there is a clear role for PtdIns(3)P homeostasis in the recruitment of RUFY3 on EL, this effect is likely to be indirect and we have therefore no evidence for a functional role of the FYVE domain. See also our answer to Reviewer 1's comment number 13.

7. Page 8 – Provide a rationale for why there is a 40% reduction in Salmonella replication in rufy3-/- RAW cells. The data does not support the conclusion that iRUFY3 plays a role in promoting intracellular replication of Salmonella in phagocytic cells.

We amended our comments to indicate the iRUFY3 is contributing to the establishment of the SCV and is therefore important for the intracellular replication of Salmonella in phagocytic cells (see also comments for reviewer 2).

8. Page 11 and Figure 9 – This attempt to place biological significance for iRUFY3 deficiency in CD11c+ cells using an E. coli/lung challenge model is flawed at several levels. The protocol is not adequately described in the Methods section and the figure captions do not describe the abbreviations adequately so it is difficult to determine which cell types are being described. At the organismal level, there was no significant difference in the morbidity or mortality between the CD11ccre rufy3fl/fl mice and control animals. The authors might consider exploring other challenge models to discover a phenotype that could provide insights into the mechanism of action of iRUFY3 in CD11c+ cells.

We have corrected the material and methods to follow-up on these comments. Our objectives were to demonstrate that iRUFY3 is important for the immune response at the organismal level and that some of the observations, that we have performed in vitro can be transposed in vivo. We feel that changing models and discover new phenotypes should be part of a new cycle of research and discovery in our laboratory and obviously the object of a new article.

9. Does iRUFY3 come up in any of the human GWAS data sets as being associated with any disease condition?

We found several human RUFY3 GWAS association with data sets covering bipolar disorders, amyotrophic lateral sclerosis and parkinson's disease, but not with obvious inflammatory conditions.

REVIEWERS' COMMENTS

Reviewer #1 (Remarks to the Author):

In light of new experiments and revisions to the manuscript text, the authors have addressed most of the comments and suggestions.

Reviewer #2 (Remarks to the Author):

The authors have satisfied most of my requests, and they provided sound and convincing arguments in their point-by-point answers. The manuscript is now very pleasant to read and, I believe, will provide many novel and relevant insights for Nat. Comm. readership. Congratulations!

I still have a few minor points that can be addressed with text and figure edits:

- The authors wrote: "it [iRUFY3] localizes to the SCV containing GFP-Salmonella, LAMP1, and the bacterial effector PipB2 (Fig. 7c and 7d)." As I mentioned in the first feedback round, the images provided do not demonstrate a clear localization of iRUFY3 at the SCV, mainly because the cells are filled with intracellular bacteria. Panel d is clearer but only a small percentage of bacteria are surrounded with an iRUFY3 signal (however, the change of iRUFY3 localization upon infection is very clear). The authors argued that they prefer to keep further investigations of the role of iRUFY3 during Salmonella infection for future studies – which is understandable as this is a minor element of the current manuscript. Yet, they should amend the text to make it match what can be concluded from the figure: iRUFY3 is found in the vicinity of some SCVs at 16 hpi. Similarly, the sentence "Thus, iRUFY3 contributes to SCV maturation" should be toned down as this is currently an assumption. The section title, "RUFY3 contributes to the formation of the Salmonella-containing vacuoles", should also be changed to focus on intracellular Salmonella replication/survival.
- The authors wrote: "Equivalent uptake of WT and replication-incompetent (DsiFA) bacteria was observed after 2h of infection in the different cells tested". The authors should indicate what the "different cells tested" are, as they don't test each of the 4 cell lines mentioned above in the text.
- In the x-axis of Fig7a-b, "WT" might designate RAW or Salmonella. To make the figure easy to interpret, the authors should precise "RAW WT + SL DsiFA"; "RAW WT + SL WT", etc.
- There are 2 black squares in Fig.7d that the authors forgot to remove.

Reviewer #3 (Remarks to the Author):

The revisions to the text are acknowledged. Despite these changes, the text lacks clarity and precision and the layout of the main and extended figures makes the paper unnecessarily difficult to read and to interpret the data. The methods section still lacks important information required for reproducibility and interpretation.

While the microscopy data provide strong observations that iRUFY3 is associated with the EL network, the conclusions that iRUFY3 is actively 'recruited' on LAMP1+ ELs and is 'bound to' ARL8b via its extended C-terminal domain is an over-interpretation of the data presented. It may well be that these precise molecular interactions are indeed taking place in RAW cells, but the resolution of the imaging approaches used and the temporal dynamics of the experimental design fall short of confirming these molecular interactions. The observations appear to be technically sound but the nature of the author's mechanistic conclusions/interpretations need to be in keeping with the fundamental limitations of the type of data presented.

While the EL work appears sound, the work to demonstrate the in vivo significance for selective deficiency of iRUFY3 in CD11c+ macrophages and DCs at the organism-level is unchanged from the original submission and thus remain plagued with the same fundamental flaws and weaknesses. The main findings remain descriptive and provide superficial observations with no substantive mechanistic insight that links selective CD11c iRUFY3 deficiency to the changes in the cellular and molecular phenotype. The IP LPS or IT E. coli challenges result in highly dynamic and complex systemic and local responses .

Point to point answers to the reviewers' comments.

We thank all the reviewers for their extremely positive comments on our manuscript and their constructive criticisms. We have now address most of their remaining concerns by introducing text amendments.

Reviewer #1 (Remarks to the Author):

In light of new experiments and revisions to the manuscript text, the authors have addressed most of the comments and suggestions.

We thank reviewer #1 for acknowledging the efforts done to follow his/her advises to improve the manuscript.

Reviewer #2 (Remarks to the Author):

The authors have satisfied most of my requests, and they provided sound and convincing arguments in their point-by-point answers. The manuscript is now very pleasant to read and, I believe, will provide many novel and relevant insights for Nat. Comm. readership. Congratulation!

We thank reviewer #2 for his laudatory appraisal and for acknowledging the efforts done to follow his/her advises to improve the manuscript.

I still have a few minor points that can be addressed with text and figure edits:

- The authors wrote: "it [iRUFY3] localizes to the SCV containing GFP-Salmonella, LAMP1, and the bacterial effector PipB2 (Fig. 7c and 7d)." As I mentioned in the first feedback round, the images provided do not demonstrate a clear localization of iRUFY3 at the SCV, mainly because the cells are filled with intracellular bacteria. Panel d is clearer but only a small percentage of bacteria are surrounded with an iRUFY3 signal (however, the change of iRUFY3 localization upon infection is very clear). The authors argued that they prefer to keep further investigations of the role of iRUFY3 during Salmonella infection for future studies – which is understandable as this is a minor element of the current manuscript. Yet, they should amend the text to make it match what can be concluded from the figure: iRUFY3 is found in the vicinity of some SCVs at 16 hpi. Similarly, the sentence "Thus, iRUFY3 contributes to SCV maturation" should be toned down as this is currently an assumption. The section title, "RUFY3 contributes to the formation of the Salmonella-containing vacuoles", should also be changed to focus on intracellular Salmonella replication/survival.

- The authors wrote: "Equivalent uptake of WT and replication-incompetent (DsifA) bacteria was observed after 2h of infection in the different cells tested". The authors should indicate what the "different cells tested" are, as they don't test each of the 4 cell lines mentioned above in the text.

- In the x-axis of Fig7a-b, "WT" might designate RAW or Salmonella. To make the figure easy to interpret, the authors should precise "RAW WT + SL DsifA"; "RAW WT + SL WT", etc.

- There are 2 black squares in Fig.7d that the authors forgot to remove.

We have included all the suggested text modifications and figures edits in the new manuscript version. The title of the section and figure now reads "RUFY3 is required for intracellular *Salmonella* replication in macrophages".

Reviewer #3 (Remarks to the Author):

The revisions to the text are acknowledged. Despite these changes, the text lacks clarity and precision and the layout of the main and extended figures makes the paper unnecessarily difficult to read and to interpret the data. The methods section still lacks important information required for reproducibility and interpretation.

We thank reviewer #3 for acknowledging the efforts done to follow his/her advises to improve the manuscript, that were also found pleasant to read by reviewer #2. We had already modified the methods section for reproducibility in the previous version of the manuscript and have tried to improve it again in this version.

While the microscopy data provide strong observations that iRUFY3 is associated with the EL network, the conclusions that iRUFY3 is actively 'recruited' on LAMP1+ ELs and is 'bound to' ARL8b via its extended C-terminal domain is an over-interpretation of the data presented. It may well be that these precise molecular interactions are indeed taking place in RAW cells, but the resolution of the imaging approaches used and the temporal dynamics of the experimental design fall short of confirming these molecular interactions. The observations appear to be technically sound but the nature of the author's mechanistic conclusions/interpretations need to be in keeping the fundamental limitations of the type of data presented.

We showed using subcellular fractionation that iRUFY3 is strongly enriched on organelles likely to be LAMP1+ ELs after LPS treatment (Fig. 3e). Moreover, *Arl8b* knock-down alters also the expression of RUFY3, suggesting that the two molecules are coupled biochemically and could form a complex. These results support therefore an active recruitment of the Arl8b/RUFY3 complex to ELs in LPS or nutrient starvation conditions. Although the direct interaction of ARL8b and iRUFY3 have been shown by 2 other laboratories, we did not formally demonstrate it in this manuscript (PLA demonstrating only a vicinity of less than 40nm). We therefore amended the manuscript in several occurrences to follow reviewer #3 advises and minimize overinterpretations of our data. In addition of several other text modifications:

we removed the sentence:

"Page 7. confirming that conversely to nRUFY3, iRUFY3 is actively recruited on LAMP1+ ELs and its extended C-terminal domain is determinant for this localization."

Modified the sentence:

"Page 8. Importantly the same experiments performed in RAW-nRUFY3 did not indicate any significant interaction of nRUFY3 with ARL8b (Fig. 4c) confirming that the neuronal isoform does not localize to ELs (Extended Data Fig. 2a), and confirming that the domain of interaction of iRUFY3 with ELs and potentially ARL8b is located within the last 200 aa residues of the protein and not within its RUN domain^{21, 22}."

For the sentence:

"Page 8. The same experiments performed in RAW-nRUFY3 did not indicate any interaction of nRUFY3 with ARL8b (Fig. 4c) confirming that the neuronal isoform does not localize to ELs (Extended Data Fig. 2a), and confirming that the domain of interaction of iRUFY3 with ELs is located within the last 200 aa residues of the protein and not within its RUN domain^{21, 22}."

Modified the sentence:

“Page 9. Thus, although its recruitment to ELs appears to be mostly mediated by ARL8b, RUFY3 may have the capacity to interact with other membrane-associated molecules if a decreased competition for ARL8b binding occurs as during silencing of the small GTPase”.

For the sentence:

“P9. Thus, although its recruitment to ELs appears to be mostly linked to that of ARL8b, RUFY3 may have the capacity to interact with other membrane-associated molecules if ARL8b levels are decreased. »

While the EL work appears sound, the work to demonstrate the in vivo significance for selective deficiency of iRUFY3 in CD11c+ macrophages and DCs at the organism-level is unchanged from the original submission and thus remain plagued with the same fundamental flaws and weaknesses. The main findings remain descriptive and provide superficial observations with no substantive mechanistic insight that links selective CD11c iRUFY3 deficiency to the changes in the cellular and molecular phenotype. The IP LPS or IT E. coli challenges result in highly dynamic and complex systemic and local responses.

As stated in our first rebuttal letter, the phenotypes observed in the CD11C-RUFY3^{fl/fl} mouse are clearly reproducible and point towards a pro-inflammatory situation in the whole animal. Importantly, we have shown in the first part of our work that RUFY3 deficiency impacts activated phagocytes migration as well as IFN-gamma sensitivity and MHC II restricted antigen presentation. Both of these observations could provide a mechanistic explanation for the phenotypes observed upon LPS-stimulation of CD11C-RUFY3^{fl/fl} mouse, as supported by the detailed splenic cell populations analysis in the different experimental conditions provided in Figure 10. We agree that at the light of these observations, the CD11C-RUFY3 flx/flx mouse model deserves further immunological investigations and work on primary DCs and alveolar macrophages will be needed in the future. We however argue that these observations in vivo support a role for RUFY3-dependent EL positioning in the control of inflammation and probably antigen presentation and are necessary to complement our in vitro findings, while orientating future research projects.